



# Analytical and modelling strategies for thermal histories from in situ (U-Th-Sm)/He data of single apatites

Ann-Kathrin Maier[2,1], Christoph Glotzbach[1], Sarah Falkowski[3,1]

[1]Department of Geosciences, University of Tübingen, 72076 Tübingen, Germany
[2]Institute of Seismology, Department of Geosciences and Geography, University of Helsinki, Helsinki, 00014, Finland
[3]School of Geographical & Earth Sciences, University of Glasgow, Glasgow, G12 8RZ, UK

*Correspondence to*: Ann-Kathrin Maier (ann-kathrin.maier@helsinki.fi)

**Abstract.** (U-Th-Sm)/He is a thermochronometric method used to reconstruct the rates and timing of geological processes. Recent developments in analytical approaches, specifically laser ablation (in situ) measurements, allow quantifying the
distribution of parent isotopes (U, Th, and, in apatites, Sm) and decay products ($^4$He) within individual mineral grains. This is particularly important to understand potential date over-dispersion, which can arise from the heterogeneous distribution of parent isotopes, and to develop thermal history modelling for single-grain (U-Th-Sm)/He techniques.

We build on previous studies and combine in situ $^4$He concentration profile measurements with parent nuclide distribution mapping in natural apatites to explore analytical and modelling strategies for single-grain thermal history reconstructions.
Specifically, we investigate the effects of laser ablation spot size, the number and location of ablation spots in a grain, and grain size on data resolution and suitability for thermal history modelling. In doing so, we introduce the calculation of $C_{aw}$, which is the concentration of parent nuclides at each ablation site weighted by alpha-particle stopping distances to account for the redistribution of $^4$He in the crystal from high-energy alpha decay. We present stacked U, Th, and Sm maps measured at different ablation depths in two apatite grains from South Germany (one with homogeneous and one with zoned parent isotope
distribution) and one apatite from the McClure Mountain Syenite age standard. Furthermore, we show in situ $^4$He profiles of the two South German apatites and inversions for thermal histories. Our results indicate that, for our study and instrument set-up, four to six spot measurements with various distances from the grain rim enable measuring an in situ $^4$He profile. We determined that the optimal spot diameter for in situ $^4$He profile measurements for apatite grains with (U-Th-Sm)/He dates as young as 16 Ma is 20-30 μm. Additionally, a six-spot in situ $^4$He profile requires a minimum grain diameter (measured
perpendicular to the c-axis) of 145 μm. Combined with information from detailed parent nuclide maps, the in situ $^4$He profiles offer a possibility to reconstruct the thermal histories of single grains, potentially including zoned and irregularly shaped crystals.

## 1 Introduction

Temperature-sensitive geologic processes, including mountain building, fault activity, landscape and sedimentary basin
evolution, and ore deposit formation can be constrained with low-temperature thermochronology techniques such as (U-Th-Sm)/He (e.g., Ehlers, 2005; McInnes et al., 2005). Due to its comparatively low nominal closure temperature of ~70 °C (e.g.,



Wolf et al., 1996, 1998; Shuster et al., 2006), apatite (U-Th-Sm)/He (AHe) is particularly well-suited for constraining the thermal history of such upper crustal processes. Fundamentally, AHe is based on the competing ingrowth and thermally activated diffusive loss of alpha-particles ($^4$He) from the radioactive decay of the uranium and thorium decay chains and

samarium in the crystal lattice. Diffusive loss occurs over a specific temperature range, the helium partial retention zone (e.g., Zeitler et al., 1987; Farley, 2002; Fitzgerald et al., 2006). Apart from apatite, other minerals incorporating significant amounts of uranium and thorium and harbouring characteristic temperature sensitivities, such as zircon, titanite, hematite and monazite, can also be used for (U-Th-Sm)/He dating (e.g., Ault et al., 2019).

The amount of helium retained in a crystal is a function of the time-temperature evolution of a rock sample and the crystal-

specific properties affecting the diffusivity, including (1) the grain size and geometry determining the diffusion domain and the alpha-particle ejection at the grain boundary, (2) the concentration of effective uranium (eU=U+0.235×Th) representative of the extent of self-irradiation-induced crystal lattice defects (i.e., radiation damage), (3) the presence of fluid and mineral inclusions and phases around the crystal contributing potential excess $^4$He, and (4) the distribution of parent nuclides (e.g., Farley et al., 1996, 2011; Reiners and Farley, 2001; Shuster et al., 2006; Vermeesch et al., 2007; Spiegel et al., 2009; Gautheron

et al., 2012; Anderson et al., 2017). A meaningful geological interpretation of (U-Th-Sm)/He dates thus requires understanding and accounting for these aspects.

Beyond that, reconstructing thermal histories from (U-Th-Sm)/He data is challenging due to the inability to constrain cooling histories solely based on a single (U-Th-Sm)/He date, as a date is non-unique regarding possible time-temperature paths (e.g., Shuster and Farley, 2004). Researchers thus developed different strategies, such as the use of crystals with varying kinetic

properties (i.e., grains of varying sizes, radiation damage, or grain fragments), the combination of different thermochronometer systems, and the analysis of samples taken along a quasi-vertical elevation profile to overcome this limitation (e.g., Reiners and Farley, 2001; Fitzgerald et al., 2006; Flowers, 2009; Flowers and Kelley, 2011; Beucher et al., 2013; Brown et al., 2013). In addition to such approaches involving multiple mineral grains, the shape of a single grain's diffusion profile, acquired through proton irradiation and subsequent stepwise degassing, is exploited in the $^4$He/$^3$He method with the rationale that a $^4$He

profile reflects the duration of active diffusion a crystal experienced and, hence, its possible thermal history (Shuster and Farley, 2004). While, for example, a more rounded profile towards the grain rim would indicate slow cooling, a uniform $^4$He distribution would be produced by faster cooling (Shuster and Farley, 2004). A heterogeneous parent radionuclide distribution in a grain may complicate the interpretation of $^4$He concentration profiles (e.g., Farley et al., 2011).

For thermal modelling, it is essential to characterise the spatial distribution of $^4$He and its parent radionuclides, and to

understand sources of possible (U-Th-Sm)/He date dispersion, such as parent nuclide zonation (e.g., Farley et al., 2011; Vermeesch et al., 2012; Danišík et al., 2017; Idleman et al., 2018; Sousa et al., 2024). The in situ technique to determine both helium and trace element content via laser-ablation promises new insights compared to more established whole-grain protocols (e.g., Gautheron et al., 2021), where the spatial relationship between parent nuclides and decay products in single grains generally remains unquantified (Boyce et al., 2006; Vermeesch et al., 2012, 2023; Danišík et al., 2017; Glotzbach and Ehlers,

2024). Not least, in situ mapping of parent nuclides and $^4$He allows the determination of a single grain's possible thermal



history. Danišík et al. (2017) demonstrated this by assessing the spatial relationship of uranium, thorium, and helium in zircons by µm-scale laser ablation inductively coupled plasma mass spectrometry (LA-ICP-MS) element mapping and conversion of their detailed 2D maps into 1D concentration profiles to then invert for a possible single-grain thermal history. Recently, Glotzbach and Ehlers (2024) suggested optimised strategies for the reconstruction of cooling histories from in situ measurements based on synthetic data modelling and the incorporation of in situ (U-Th-Sm)/He adapted helium production-ejection-diffusion models. They suggested using either in situ measurements of multiple grains of varying size or eU, similar to the whole-grain method, or multiple in situ spot measurements along a core-to-rim profile in a single grain. However, they did not test their strategies on natural samples.

In this study, we expand on the work by Danišík et al. (2017) and Glotzbach and Ehlers (2024) and test whether it is possible to obtain reliable single-grain helium concentration profiles from in situ $^4$He measurements and combine them with parent nuclide maps for cooling history inversion. We explore analytical and thermal modelling strategies for best results using natural samples from South Germany with homogeneous and heterogeneous radionuclide distributions and a large and clear apatite from the McClure Mountain Syenite (Colorado, USA). Specifically, we examine the number of ablation spots needed to retrieve an interpretable $^4$He profile and evaluate limitations on grain size and ablation spot location and size.

## 2 Methods

This section presents our analytical workflow (Fig. 1), including the $^4$He profile and parent nuclide measurement protocols, data visualisation, and thermal history modelling strategy. As detailed descriptions of laser-ablation in situ (U-Th-Sm)/He analyses are provided elsewhere (e.g., Boyce et al., 2006; Horne et al., 2016), we focus on the specifics of our procedure.

### 2.1 Samples and sample preparation

We analysed apatites extracted from different lithologies in South Germany and the McClure Mountain Syenite age standard (523.5±1.5 Ma, Schoene & Bowring, 2006) (Table 1). Datable crystals were selected based on the criteria for whole-grain analyses, i.e., no visible inclusions, fractures, defects, and rounded or broken edges and tips, and a diameter larger than 60 µm (e.g., Farley, 2002), and photographed parallel and perpendicular to the c-axis following the *3D-He* protocol of Glotzbach et al. (2019) to record the grain geometry information needed for thermal history modelling. Afterwards, the grains were embedded in a Teflon sheet with their c-axis parallel to the mount surface, ground down, and polished to approximately half-thickness. The amount of material removed was tracked with reference glass beads of known diameter, as described by Pickering et al. (2020). Imaging the mount with a tabletop scanning electron microscope (SEM) before laser ablation analysis did not reveal internal zonation in any of the chosen crystals (SEM images are shown in Section 3.2).





**Table 1: Sample information**

| Sample | Lithology and crystallisation age | Location | Longitude | Latitude |
|---|---|---|---|---|
| Apatite-URG | Foiditic tuff (16.75±0.84 Ma, Binder et al. 2023) | Herbolzheim (Upper Rhine Graben) | 7.779325 | 48.2319861 |
| Apatite-BaF | Biotite-rich coarse granite (Variscan) | Prenning (Bavarian Forest) | 12.939167 | 49.016389 |
| Apatite-McClure | Hornblende-biotite syenite (523.5±1.5 Ma, Schoene & Bowring, 2006) | Wet Mountains (Colorado) | -105.483333[a] | 38.35[a] |

[a] These are approximate coordinates based on the original sample locality reported in Alexander et al. (1978).

## 2.2 In situ helium profile measurements

We acquired $^4$He concentration profiles from multiple in situ $^4$He spot measurements along several c-axis perpendicular and one c-axis parallel traverses through single crystals (Fig. 1) to evaluate the influence of the measurement location, the consistency of the results, and potential influences of parent nuclide heterogeneities. This resulted in 28–38 individual ablation sites per crystal (Table 2).

The in situ $^4$He measurements were conducted with a RESOchron system (Applied Spectra) consisting of a He-line and an
excimer laser at the University of Tübingen, Germany. All analysed grains were ablated for 8 s with a laser pulse frequency of 10 Hz and a laser fluence of 2 J cm$^{-2}$. The laser ablation spots, sized 10 to 30 µm in diameter, were spaced 3 to 5 µm apart to avoid signal smearing and mixing (Fox et al., 2017). The laser spot sizes were chosen individually for each grain and set as small as possible to ensure acceptable He-signals three standard deviations above the blank level (Table 2). Line blanks were recorded regularly in the ablation sequence and ranged from 0.0003 to 0.0005 ncc. Blank correction, Q-shot interpolation to
account for instrumental drift, and $^4$He content calculation were performed using in-house software.

For successful $^4$He measurements, standard deviations after blank correction ranged from 6–15 % (Table 2). After $^4$He measurements, the surface topography of the analysed grains was imaged using a confocal laser-scanning microscope (Zeiss LSM 900) to determine the ablation pit dimensions. Based thereon, the ablation pit volumes were obtained in the Zeiss Confomap software and used to calculate pit-volume normalised $^4$He concentrations. For Apatite-BaF and Apatite-McClure,
we used the mean pit volume to calculate the $^4$He concentrations due to a large spread in measured pit volumes (see Section 4.2 for limitations of pit volume measurements). Detailed pit volumes and depths for individual ablation spots are listed in Table 3. Mean pit volumes in the analysed apatites ranged from 310 µm$^3$ ± 10 % to 4240 µm$^3$ ± 4 % (1 SD; Table 2).





## 2.3 Parent nuclide mapping

We performed detailed parent nuclide mapping to garner the necessary information for thermal modelling and to assess the
possible influence of U, Th, and Sm heterogeneities on the measured [4]He distribution following the example of Danišík et al.
(2017). The LA-ICP-MS measurements were conducted on an evenly spaced grid of non-overlapping spots (Fox et al., 2017)
across smoothly re-polished grain surfaces, following ablation for [4]He measurements, with a spot diameter of 24 µm and a
spot depth of approximately 24 µm. The ablation time was 12 s with a laser fluence of 3 J cm[-2] and a pulse frequency of 20
Hz. We used NIST612 and Durango as reference material for apatite in the standard bracketing approach to estimate trace
element concentrations (Paton et al., 2010). Removal of outliers (per default all measured counts per second (CPS) more than
three standard deviations away from a running mean), background correction, and trace element concentration calculation
were performed with an in-house MATLAB app (ESD-U-Pb).

To construct stacked 2D maps of parent nuclide distributions from deep ablation spots on just one internal surface, we used
the "downhole" time-resolved measurements and the approximate ablation time-depth relationship. The latter was determined
by measuring pit depths corresponding to 2–18 s ablation times in spare apatite grains of the same samples. The resulting time-
depth relationship was approximately linear, with an ablation rate of ~2 µm s[-1].

Finally, we computed sub-ablation-spot resolution U, Th, and Sm distribution maps from neighbouring spot measurements
using the regularised linear least squares MATLAB code by Fox et al. (2017). Such a regularised inversion requires balancing
model smoothness and complexity by choosing an adequate regularisation parameter or smoothness constraint $\lambda$. The
smoothness constraint controls the influence the penalty term for model complexity has on the inversion result. A too-large
smoothness constraint leads to retrieving parent nuclide maps that are too smooth and do not capture the underlying true
concentration variations. Conversely, if the regularisation parameter is too small, the inversion solution will be dominated by
data errors, and every small concentration change (noise) will be matched. Following Fox et al. (2017), we chose the
smoothness constraint based on qualitative information from SEM and the L-curve criterion (e.g., Hansen and O'Leary, 1993).
The L-curve is a log-log plot of the residual against the norm of the regularised solution parameterised by the smoothness
constraint, which is often L-shaped. The idea is to choose the smoothness constraint that corresponds to the corner of the "L".
In this way, we computed 2D parent nuclide distribution maps with a resolution of 10x10 µm (Apatite-URG) and 5x5 µm
(Apatite-BaF, Apatite-McClure) for each recorded laser penetration depth. We stacked those map slices to display a pseudo-
3D section through the analysed part of the grain (Section 3.3).







**Table 2: ⁴He- and trace element measurement details**

| Sample | Grain radius [µm] | ⁴He | | | | | Trace elements | |
|---|---|---|---|---|---|---|---|---|
| | | Number of ablation spots | Ablation spot diameter [µm] | Mean spot depth [µm] | Mean ablation pit volume ± 1 SD [µm] | 1 SD ⁴He[a] [%] | Number of ablation spots | Ablation spot diameter, depth [µm] |
| Apatite-URG | 175 | 32 | 30 | 8.1 | 4240 ± 170 | <15 | 356 | 24, 24 |
| Apatite-BaF | 89 | 28 | 20 | 7.9 | 1550 ± 140 | <6 | 90 | 24, 24 |
| Apatite-McClure | 75 | 38 | 10 | 9.7 | 310 ± 30 | >40 | 84 | 24, 24 |

[a] This is the ⁴He measurement uncertainty after blank correction.

SD stands for standard deviation.

## 2.4 Alpha-stopping distance weighted parent nuclide concentration $C_{aw}$

As the in situ ⁴He (spots along profiles) and parent nuclide measurements (spots for 2D maps) do not correspond to the same location in the grain in our procedure (Fig. 1), we had to match the separate U, Th, Sm and ⁴He measurements for thermal modelling. For this purpose, we determined an alpha-stopping distance weighted parent nuclide concentration ($C_{aw}$, **C**oncentration **a**lpha-**w**eighted) at each helium ablation site. Although other options to make information from 2D parent nuclide maps usable for thermal modelling already exist, for example, calculating 1D equivalent-sphere geometry concentration profiles (e.g., Farley et al., 2011; Danišík et al., 2017), we introduce this alpha-stopping distance weighted parent nuclide concentration because it allows us to account for the emission and redistribution of ⁴He (alpha particles) from the decay site during high-energy decay. Since ⁴He measured in a spot is the result of the parent nuclides that surround it within the alpha-stopping distance reach (e.g., Farley et al., 2010), we determined $C_{aw}$ from the distribution of parent nuclides in each ⁴He spot's periphery as follows. First, we calculated the mean U, Th, and Sm concentrations around the centre point of each ⁴He measurement spot for spheres with radii corresponding to all possible alpha-stopping distances (between ~6–40 µm, Ketcham et al., 2011). Then, we summed the mean parent nuclide concentration for each stopping distance weighted by its contribution to ⁴He production (Eq. 1).

$$C_{aw} = \sum_j^m f_j \frac{\sum_{i=1}^n c_{i,j}}{n} \text{ (Eq. 1)}$$

In Equation 1, $c$ is the parent nuclide concentration within a certain stopping distance, $n$ is the number of concentration measurements, $m$ is the number of stopping distances, and $f$ is the weight for the contribution to the production of ⁴He.

The $C_{aw}$ calculation is based on the available 3D information on the parent nuclide distribution and is, hence, constrained by the resolution and accuracy of the measured parent nuclide maps. It thus depends on the number of mapped grain slices, the accuracy of the ablation time-depth relationship (Section 2.3), and the fact that information of the top half of the grain is inevitably lost from grinding it down. Due to the latter, we made the following simplifying assumptions. (1) Grains are mirror-symmetrical about the polished internal grain surface, (2) helium and trace elements were measured in the same plane, and (3)



where there is a lack of 3D data, we assume the same concentration as for the closest measurement (interpolation) point. Finally, we chose not to calculate $C_{aw}$ for $^4$He ablation spots with centres <40 µm to the grain rim (maximum alpha-stopping distance, Ketcham et al., 2011) because at the grain rim, $^4$He is not only redistributed, but can also be ejected and lost or implanted (e.g., Farley et al., 1996).

## 2.5 Thermal history modelling

The shape of a $^4$He concentration profile in a grain is largely a function of the duration of active diffusion and, thus, thermal history (Shuster and Farley, 2004). We can, therefore, reconstruct thermal histories by inverting the in situ $^4$He profile measurements and the corresponding alpha-stopping distance weighted parent nuclide concentrations ($C_{aw}$). We applied the modelling technique outlined by Glotzbach and Ehlers (2024), which allows predicting the $^4$He concentrations at specific locations in a grain, assuming a cylindrical grain geometry and considering the full range of alpha-stopping distances. Glotzbach and Ehlers's (2024) MATLAB code is an adjustment of the radiation damage accumulation and annealing models (RDAAM, Flowers et al. (2009), and ZrDAAM, Guenthner et al. (2013)) implemented in HeFTy (Ketcham, 2005; Ketcham et al., 2018; Ketcham, 2024). The RDAAM (apatite) and ZrDAAM (zircon) models treat $^4$He diffusion in a grain as a function of accumulated self-irradiation damage and related diffusivity variations over the grains' thermal evolution (Flowers et al., 2009; Guenthner et al., 2013). Using the approach of Glotzbach and Ehlers (2024), helium production and diffusion was calculated for 5000 (Apatite-URG) and 10000 (Apatite-BaF) random time-temperature paths based on the horizontal and vertical distance of a $^4$He ablation spot centre to the grain rims, the $^4$He pit depth, the grain radius, and the U, Th, Sm, and $^4$He concentrations. Each path's goodness of fit (GOF) was evaluated as in HeFTy, where a GOF of 0.05 corresponds to acceptable time-temperature paths passing the 95% confidence test and a GOF of 0.5 (statistical precision limit) to good paths (Ketcham, 2005; Ketcham, 2024).

The paths with the highest GOF were selected to forward-model the corresponding $^4$He profiles. The misfit $m$ between modelled and measured $^4$He profiles was calculated as

$$m = \sqrt{\sum_{i=1}^{n} \frac{r_i^2}{\sigma_i^2}} \qquad \text{(Eq. 2)}$$

with $r_i$ being the residual between measured and modelled concentration at the $i^{th}$ $^4$He spot and $\sigma_i$ being the measurement uncertainty, to narrow down the possible time-temperature paths. This way, a limited number of plausible cooling histories is computed for a grain, which can be interpreted in the geological context.





**1. Grain geometry:**
photo-
micrographs of
select grains

**2. Grain mounting**
grinding and
polishing

Teflon

grain

**3. He measurements**
along profiles &
pit volume
determination

laser
ablation

**He pit
measurement**

**⁴He profile**

Q-MS

confocal laser-scanning
microscope

[⁴He]

-        0        +

**Distance from grain
centre [μm]**

**4. Polishing**
to create an even
surface

**Parent nuclide maps**

**5. U-Th-Sm-
measurements**
with LA-ICP-MS
& interpolation to
2D maps

laser
ablation

interpolation

LA-ICP-MS

z [μm]

x [μm]

**6. Inversion of in-
situ (U-Th-Sm)/He
data**
for cooling histories

**Cooling history**

T [°C]

**Time [Ma]**

**Figure 1:** Schematic depiction of the analytical protocol for in situ ⁴He profile measurements and parent nuclide mapping to reconstruct thermal histories of single grains. Q-MS: quadrupole mass spectrometer; LA-ICP-MS: laser ablation inductively coupled plasma mass spectrometry.


## 3 Results

### 3.1 In situ $^4He$ concentrations and uncertainties

The grains examined in this study span a broad range of $^4He$ concentrations and associated uncertainties, highlighting differences in parent nuclide concentration and cooling history. In situ $^4He$ concentrations range from 1.7E15 ± 2.2E14 at $g^{-1}$ to 2.1E15 ± 3.0E14 at $g^{-1}$ for Apatite-URG and 1.1E16 ± 1.2E15 at $g^{-1}$ to 2.4E16 ± 2.4E15 at $g^{-1}$ for Apatite-BaF. Corresponding uncertainties after blank correction and pit volume determination are <15% and <10%, respectively (Table 2). The Apatite-McClure sample with $^4He$ concentrations of 2.8E15 ± 5.0E15 at $g^{-1}$ to 8.5E15 ± 2.5E15 at $g^{-1}$ has a comparatively high uncertainty of >40% after blank correction.

**Table 3: $^4He$ and alpha-stopping distance weighted parent nuclides concentrations ($C_{aw}$)**

| Spot | Pit volume [$\mu m^3$] | Pit depth [µm] | $^4He$ [at $g^{-1}$] | $^4He$ SD [at $g^{-1}$] | $^{238}U$ $C_{aw}$ ± 1SD [ppm] [a] | $^{232}Th$ $C_{aw}$ ± 1SD [ppm] [a] | $^{147}Sm$ $C_{aw}$ ± 1SD [ppm] [a] | Distance to grain boundary [µm] [b] | in situ AHe date ± 1SD [Ma] [c] |
|---|---|---|---|---|---|---|---|---|---|
| Ap-URG_1 | 4011 | 8.8 | 2.13E+15 | 1.93E+14 | 7.4 ± 1.0 | 107 ± 10 | 238 ± 33 | 59 | 20.0 ± 2.3 |
| Ap-URG_2 | 4121 | 8.9 | 1.95E+15 | 1.70E+14 | 8.1 ± 0.8 | 109 ± 10 | 230 ± 49 | 98 | 17.7 ± 1.9 |
| Ap-URG_3 | 4215 | 8.9 | 1.84E+15 | 2.08E+14 | 7.9 ± 0.8 | 107 ± 9 | 136 ± 19 | 135 | 17.1 ± 2.1 |
| Ap-URG_4 | 4146 | 9.1 | 1.98E+15 | 3.15E+14 | 8.3 ± 1.4 | 116 ± 19 | 217 ± 27 | 133 | 17.3 ± 3.7 |
| Ap-URG_5 | 3995 | 8.2 | 2.05E+15 | 2.70E+14 | 7.1 ± 0.7 | 98 ± 9 | 132 ± 10 | 93 | 20.9 ± 3.1 |
| Ap-URG_6 | 4150 | 8.6 | 2.02E+15 | 3.85E+14 | 7.4 ± 1.2 | 107 ± 16 | 259 ± 27 | 119 | 19.1 ± 4.2 |
| Ap-URG_7 | 4324 | 8.5 | 1.70E+15 | 2.69E+14 | 8.3 ± 0.9 | 113 ± 11 | 160 ± 11 | 158 | 15.2 ± 2.5 |
| Ap-URG_8 | 4442 | 8.9 | 1.74E+15 | 2.73E+14 | 8.2 ± 1.5 | 118 ± 23 | 146± 8 | 117 | 15.3 ± 3.5 |
| Ap-URG_9 | 4075 | 8.5 | 1.84E+15 | 2.62E+14 | 7.8 ± 1.3 | 116 ± 25 | 166 ± 11 | 79 | 16.7 ± 4.0 |
| Ap-URG_12 | 4420 | 8.2 | 1.95E+15 | 1.34E+14 | 9.1 ± 1.8 | 117 ± 14 | 123 ± 15 | 164 | 16.3 ± 2.0 |
| Ap-URG_15 | 4295 | 8.1 | 1.92E+15 | 2.86E+14 | 7.5 ± 0.9 | 107 ± 12 | 140 ± 26 | 175 | 18.1 ± 3.0 |
| Ap-URG_16 | 4217 | 7.2 | 1.78E+15 | 2.29E+14 | 8.0 ± 1.4 | 108 ± 16 | 199 ± 71 | 174 | 16.0 ± 2.8 |
| Ap-URG_17 | 4260 | 8.5 | 1.85E+15 | 2.63E+14 | 8.1 ± 0.9 | 111 ± 14 | 151 ± 21 | 173 | 16.7 ± 2.9 |
| Ap-URG_18 | 3896 | 8.1 | 2.00E+15 | 3.33E+14 | 8.0 ± 1.2 | 108 ± 16 | 184 ± 5 | 169 | 18.5 ± 3.8 |
| Ap-URG_19 | 4390 | 8.1 | 1.82E+15 | 2.98E+14 | 7.6 ± 1.1 | 101 ± 15 | 149 ± 9 | 165 | 17.9 ± 3.7 |



| | | | | | | | | | |
|---|---|---|---|---|---|---|---|---|---|
| Ap-URG_20 | 4287 | 8.0 | 1.84E+15 | 3.10E+14 | 7.5 ± 0.9 | 99 ± 10 | 122 ± 10 | 161 | 18.2 ± 3.3 |
| Ap-URG_21 | 4265 | 7.9 | 1.74E+15 | 2.11E+14 | 8.0 ± 1.2 | 106 ± 17 | 140 ± 23 | 156 | 16.2 ± 3.0 |
| Ap-URG_22 | 4526 | 8.2 | 1.69E+15 | 2.21E+14 | 8.0 ± 1.0 | 108 ± 13 | 199 ± 68 | 153 | 15.5 ± 2.3 |
| Ap-URG_28 | 4225 | 8.0 | 2.04E+15 | 2.48E+14 | 7.9 ± 1.3 | 113 ± 15 | 199 ± 7 | 48 | 18.3 ± 2.9 |
| Ap-URG_29 | 4589 | 7.8 | 1.85E+15 | 2.16E+14 | 9.5 ± 1.7 | 114 ± 19 | 165 ± 6 | 86 | 15.9 ± 2.8 |
| Ap-URG_30 | 4373 | 9.3 | 1.89E+15 | 2.22E+14 | 10.3 ± 1.9 | 123 ± 18 | 169 ± 36 | 126 | 14.9 ± 2.5 |
| Ap-URG_31 | 4203 | 6.9 | 1.92E+15 | 2.37E+14 | 7.8 ± 0.9 | 106 ± 11 | 153 ± 4 | 148 | 17.9 ± 2.7 |
| Ap-URG_32 | 4294 | 6.9 | 1.87E+15 | 2.77E+14 | 7.4 ± 0.8 | 104 ± 10 | 154 ± 4 | 108 | 17.8 ± 3.0 |
| Ap-BaF_1 | 1418 | 7.5 | 1.26E+16 | 1.34E+15 | - | - | - | 42 | - |
| Ap-BaF_2 | 1387 | 7.5 | 1.56E+16 | 1.67E+15 | 43 ± 6 | 15 ± 4 | 516 ± 101 | 66 | 101.35 ± 17.03 |
| Ap-BaF_3 | 1489 | 7.4 | 1.86E+16 | 1.86E+15 | 46 ± 5 | 20 ± 2 | 703 ± 182 | 60 | 111.44 ± 15.90 |
| Ap-BaF_4 | 1479 | 7.5 | 1.48E+16 | 1.54E+15 | 30 ± 5 | - | 595 ± 191 | 35 | - |
| Ap-BaF_7 | 1796 | 9.5 | 1.90E+16 | 1.88E+15 | 50 ± 3 | 19 ± 2 | 606 ± 56 | 84 | 104.66 ± 10.85 |
| Ap-BaF_9 | 1731 | 7.5 | 1.67E+16 | 1.78E+15 | 52 ± 4 | 22 ± 4 | 709 ± 36 | 86 | 88.01 ± 11.93 |
| Ap-BaF_11 | 1418 | 7.0 | 1.64E+16 | 1.85E+15 | 50 ± 4 | 22 ± 4 | 589 ± 79 | 87 | 90.45 ± 11.56 |
| Ap-BaF_14 | 1566 | 7.5 | 1.82E+16 | 1.72E+15 | 60 ± 4 | 25 ± 2 | 614 ± 93 | 88 | 83.22 ± 9.55 |
| Ap-BaF_17 | 1621 | 9.8 | 1.54E+16 | 1.65E+15 | - | - | - | 38 | - |
| Ap-BaF_18 | 1603 | 9.7 | 2.24E+16 | 2.12E+15 | 32 ± 6 | 16 ± 2 | 496 ± 31 | 64 | 125.81 ± 15.75 |
| Ap-BaF_19 | 1806 | 8.1 | 2.18E+16 | 2.30E+15 | 50 ± 6 | 23 ± 4 | 701 ± 69 | 60 | 126.81 ± 15.75 |
| Ap-BaF_20 | 1757 | 8.3 | 1.72E+16 | 1.92E+15 | (29 ± 5) | (11 ± 3) | (459 ± 38) | 33 | (162.25 ± 28.95) |
| Ap-BaF_21 | 1604 | 7.9 | 1.07E+16 | 1.17E+15 | - | - | - | 10 | - |
| Ap-BaF_22 | 1448 | 8.1 | 1.77E+16 | 1.87E+15 | 31 ± 7 | - | 594 ± 186 | 36 | - |
| Ap-BaF_23 | 1488 | 7.1 | 2.36E+16 | 2.35E+15 | 50 ± 10 | 21 ± 5 | 664 ± 152 | 62 | 133.29 ± 27.07 |
| Ap-BaF_24 | 1445 | 7.1 | 1.59E+16 | 1.62E+15 | 50 ± 5 | 17 ± 4 | 518 ± 23 | 61 | 90.18 ± 12.93 |
| Ap-BaF_25 | 1409 | 6.9 | 1.10E+16 | 1.04E+15 | - | - | - | 34 | - |
| Ap-BaF_26 | 1434 | 8.4 | 2.06E+16 | 1.98E+15 | 48 ± 6 | 22 ± 3 | 862 ± 137 | 64 | 114.88 ± 17.87 |
| Ap-BaF_27 | 1638 | 8.4 | 1.73E+16 | 2.03E+15 | (35 ± 7) | (14 ± 4) | (712 ± 193) | 38 | (138.37 ± 31.03) |
| Ap-BaF_28 | 1628 | 7.8 | 1.07E+16 | 1.50E+15 | - | - | - | 14 | - |



[a] For Ap-BaF, the alpha-stopping distance weighted parent nuclide concentrations ($C_{aw}$; see Section 2.4) listed were calculated based on the interpolated 5x5 μm parent nuclide maps. If the distance of the [4]He ablation spot to the grain boundary on the interpolated map is less than the maximum alpha-stopping distance for the specific element, $C_{aw}$ is not calculated (e.g. Ap-BaF_4). Note that locating the [4]He spots on

the parent nuclide map is subject to uncertainty, especially for non-straight grain boundaries. The undulating grain boundaries of Apatite-BaF are not accurately replicated on the interpolated map, leading to a discrepancy between the true grain boundary and the grain boundary as drawn in the interpolation. Thus, the $C_{aw}$ calculation for spots close to the grain rim needs to be treated with caution. Where the interpolation adds area to the grain, $C_{aw}$ values are reported in round brackets. Where the interpolated grain extent is smaller than the true grain, no $C_{aw}$ is calculated, even though the [4]He spot's distance from the true grain boundary would permit it (e.g., Ap-BaF_1). We did not

include affected spots for either case in the thermal modelling.

[b] c-axis orthogonal distance from the He-measurement spot centre to the nearest grain rim.

[c] AHe is apatite (U-Th-Sm)/He.

SD is standard deviation.

### 3.2 In situ measured helium profiles

The [4]He concentration profiles measured perpendicular to the crystallographic c-axis in Apatite-URG and Apatite-BaF depict two distinct [4]He patterns (Fig. 2). The three [4]He profiles acquired in Apatite-URG are indistinguishable within error and display an overall flat shape. Two of the three profiles (Ap-URG-P1 and Ap-URG-P2) may show a subtle trend of higher [4]He concentrations towards the grain rim (Fig. 2a). In contrast, the four Apatite-BaF [4]He profiles are concave-down with a significantly higher [4]He concentration near the grain centre and lower concentrations at the rims (Fig. 2b). The profiles agree

within measurement error, except for Ap-BaF-P3, which displays significantly higher [4]He concentrations in one half of the grain compared to the other profiles. Notably, peak [4]He concentrations for Ap-BaF-P2, Ap-BaF-P3 and Ap-BaF-P4 were measured c. 30 μm off-centre. We did not analyse the profiles of Apatite-McClure due to high uncertainties in the [4]He measurements (Section 3.1), limiting their meaningfulness. The [4]He measurement details for Apatite-McClure are listed in Table B1.



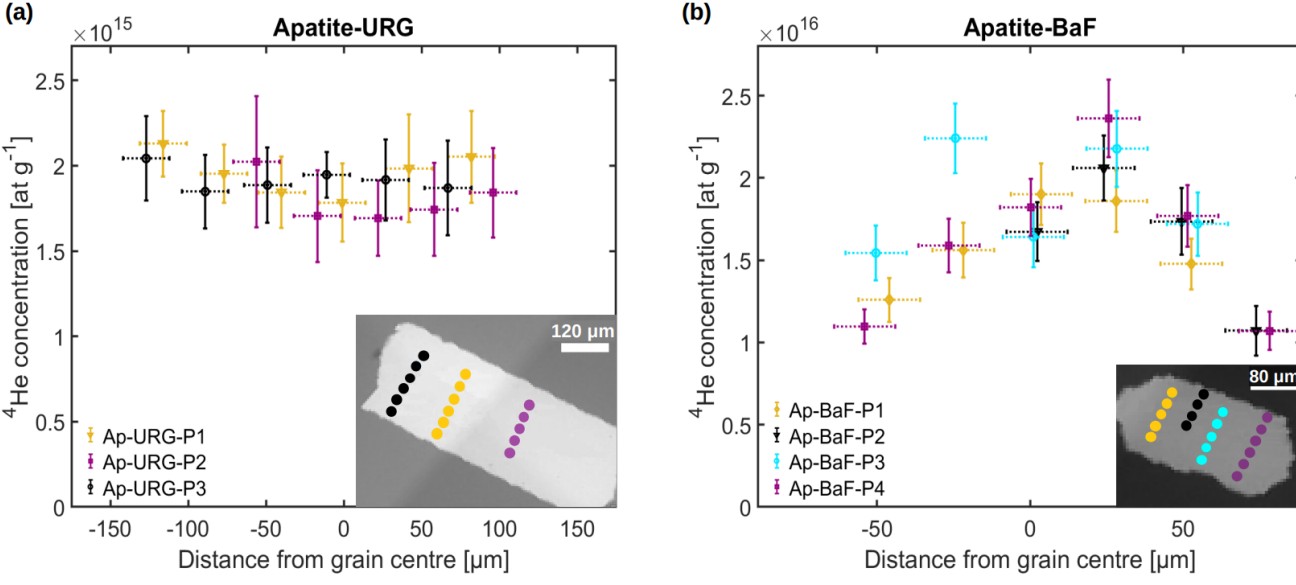

**Figure 2:** Measured in situ [4]He concentrations along c-axis-perpendicular rim-to-rim profiles in Apatite-URG (a) and Apatite-BaF (b). The coloured spots in the SEM images indicate the location of the corresponding [4]He measurements in each grain. The laser spot diameter was 30 µm for measurements in Apatite-URG and 20 µm for measurements in Apatite-BaF, indicated by the dashed horizontal error bars.

**3.3 Spatial variations in parent nuclide concentrations**

Trace element mapping offers insight into the relationship between measured [4]He profiles and parent nuclide distribution. Figure 3 shows the uppermost U, Th, and Sm maps of Apatite-URG, Apatite-BaF, and Apatite-McClure, overlaid with alpha-stopping distance weighted parent nuclide concentrations $C_{aw}$ for the [4]He ablation spots. In addition, Figure 4 displays all interpolated map slices of Apatite-BaF. Supplementary Figures A1 and A2 show all map slices for Apatite-URG and Apatite-McClure.

Apatite-McClure displays minor internal variation and comparatively low [238]U and [232]Th concentrations of 2–16 ppm and 11–25 ppm, respectively (Fig. 3a, b). As an exception, [232]Th is locally enriched at the grain rim and tip (Fig. 3b). [147]Sm (~60–250 ppm) shows higher concentrations in the core than the rim (Fig. 3c).

Similarly, Apatite-URG shows only slight variability in the [238]U concentrations (5–17 ppm), except for enriched grain rims and tips (Fig. 3e). [232]Th and [147]Sm span larger concentration ranges (86–234 ppm and 20–310 ppm, respectively) but do not
show discernible zonation patterns in either map slice (Fig. 3f, g). For each element, $C_{aw}$ does not deviate significantly from the concentrations seen in the uppermost parent nuclide map slice (Fig. 3e–g).

In contrast, Apatite-BaF has a heterogeneous parent nuclide distribution, with overall depth-consistent zonation in the [238]U (19–62 ppm), [232]Th (4–29 ppm), and [147]Sm (124–609 ppm) concentrations (Fig. 4). One side of the grain is enriched in parent nuclides compared to the other (Fig. 3i–l, Fig. 4). This matches the shapes of the measured [4]He concentration profiles that also
display [4]He enrichment in one half of the grain compared to the other. While $C_{aw}$ at each [4]He spot match the element distribution




patterns of the uppermost map slice, $^{238}$U $C_{aw}$, $^{232}$Th $C_{aw}$ and $^{147}$Sm $C_{aw}$ are overall slightly lower than in the uppermost map slice (Fig. 3i–k).

**Figure 3**: Interpolated parent nuclide (uppermost map slice) and eU maps (averaged over all slices) of the Apatite-McClure (a–d), Apatite-URG (e–h) and Apatite-BaF (i–l) grains. The smoothness constraints (see Section 2.3) for Apatite-McClure were λ=0.3 (U, Th) and λ=0.1 (Sm), for Apatite-URG λ=0.1 (U, Th) and 0.01 (Sm), and for Apatite-BaF λ=0.175 (U, Th) and λ=0.01 (Sm). Circles represent ablation spots for $^4$He. Their size reflects the laser spot size, and colours reflect the calculated alpha-stopping-distance weighted parent nuclide concentration ($C_{aw}$) (upper three rows) and the calculated in situ date based on $C_{aw}$ and $^4$He concentration (h, l). Spots for which $C_{aw}$ was not calculated are not displayed. For Apatite-McClure, in situ dates were not calculated due to very high $^4$He measurement uncertainties.



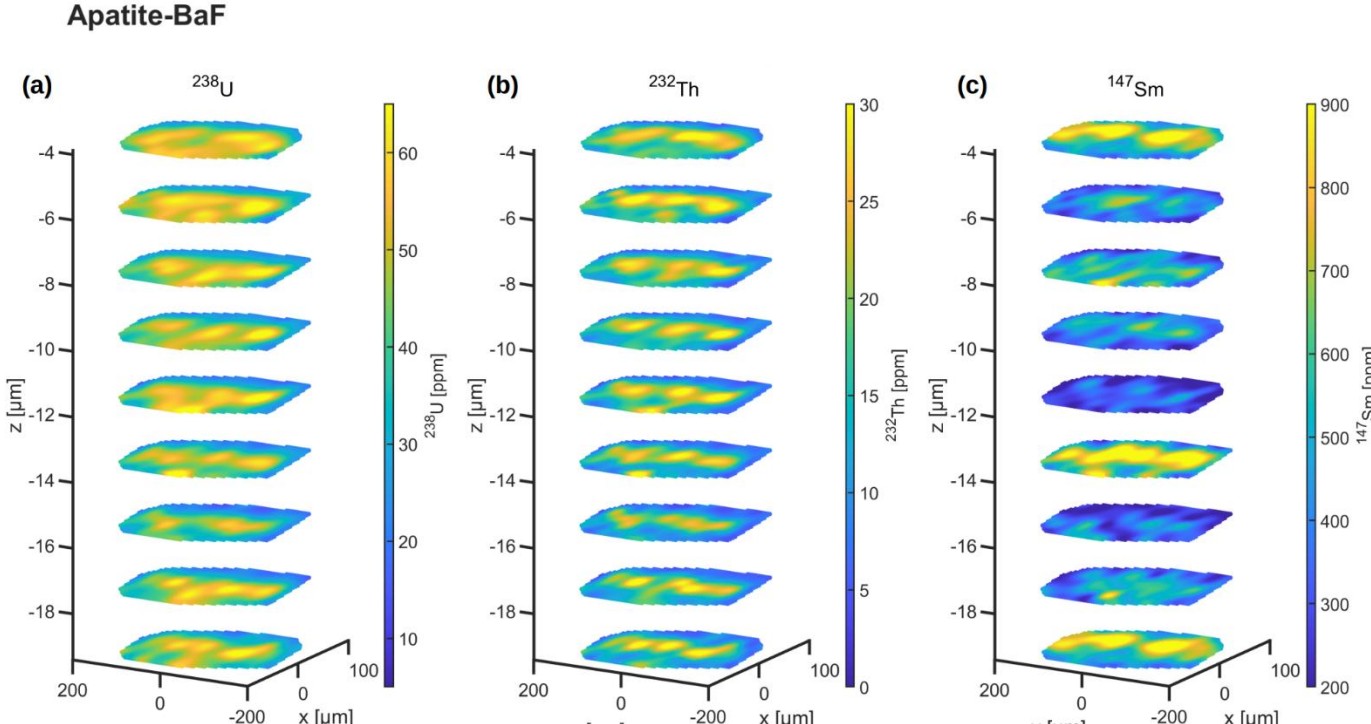

**Figure 4:** Interpolated parent nuclide distribution maps (5x5 µm horizontal resolution) of Apatite-BaF. Vertically, the parent nuclide concentrations were mapped approximately every 2 µm for a 20 µm deep section through the grain (parallel to the c-axis). The uppermost slice mapped at 2 µm depth is not displayed due to a large number of outlier measurements (Section 2.3). Parent nuclide maps were interpolated with a smoothness constraint (see Section 2.3) of λ=0.175 for the $^{238}$U (a) and $^{232}$Th (b) maps and λ=0.01 for the $^{147}$Sm maps (c).

### 3.4 Spatial variation of in situ dates

In situ AHe dates are calculated from the $^4$He concentration and $C_{aw}$ and vary with the spot location in the grain (Fig. 3h, l). In Apatite-URG, the in situ dates are the same within error, ranging from $15.2 \pm 2.5$ to $20.9 \pm 3.1$ Ma (1 SD). There is a trend of older in situ AHe dates closer to the grain rim, but a spatial correlation between the date pattern and eU is not evident (Fig. 3h). The weighted mean in situ AHe date of $17.2 \pm 1.6$ Ma is within the apatite U-Pb date of $16.75 \pm 0.84$ Ma determined by Binder et al. (2023) for this sample.

The in situ AHe dates in Apatite-BaF show a larger range ($83.22 \pm 9.55$ Ma to $162.25 \pm 28.95$ Ma, with a weighted mean date of $98.3 \pm 41.8$ Ma) and tend to be older towards the grain rims. Except for two anomalously old dates of spots closest to the grain boundary (Fig. 3l), in situ dates with a similar distance to the grain rim agree within measurement uncertainty. It appears that the youngest in situ dates are closest to the grain centre and in areas of the highest eU.





### 3.5 Cooling histories of two natural apatite crystals

### 3.5.1 Thermal histories from in situ helium profiles

In situ [4]He profiles and their corresponding $C_{aw}$ can be inverted for cooling history reconstructions of single grains, which we
tested for grains Apatite-URG (homogeneous) and Apatite-BaF (zoned).

We inverted the three [4]He profiles measured in Apatite-URG for time-temperature paths with the present-day mean annual temperature of 10 °C for Germany (German Weather Service DWD) as an endpoint constraint and allowing a deviation of ± 5 °C. The model starting point was 20 Ma and 550 °C based on the independently determined apatite U-Pb date of 16.75 ± 0.84 Ma (Binder et al., 2023). Using these two constraints resulted in models with a large number of acceptable paths (i.e.,
GOF >5%) for all input [4]He profiles, but no good paths (i.e., GOF >50%) were retrieved (Fig. 5). Further, the [4]He profiles, forward-modelled based on the acceptable paths, align with the measured [4]He profiles within measurement uncertainty. The best-fitting cooling paths have misfits (Eq. 2, cf., Section 2.5) of m=1.37 (Ap-URG-P1), m=0.84 (Ap-URG-P2) and m=1.77 (Ap-URG-P3). All [4]He profile inversions and the corresponding best-fit forward models (Fig. 5a-c) indicate rapid cooling through the He PRZ between 15 and 20 Ma, which is both compatible with the volcanic nature of the sample (tuff) and the
timing of magmatism inferred for the southern Upper Rhine Graben (Binder et al., 2023).

The in situ [4]He profile inversion for zoned Apatite-BaF only produced acceptable time-temperature paths for one of the four measured [4]He profiles (Ap-BaF-P1) (Fig. 6). Note that we only included [4]He spots for which $C_{aw}$ could be calculated (Table 3) in the inverse modelling. We used the same endpoint constraint for the time-temperature paths as for Apatite-URG, setting the temperature at 10 ± 5 °C for t=0 Ma. The starting point was set to a temperature of 570 °C at a time of 320 Ma, based on
the weighted mean apatite U-Pb date derived from trace element measurements in Apatite-BaF. Based on a study conducted near the sample location of Apatite-BaF in the Bavarian Forest, we explored a cooling-only scenario (scenario 1) with the above start- and endpoint constraints (Fig. 6a, c) and an exhumation-and-reheating scenario (scenario 2, Fig. 6b, d) (Vamvaka et al., 2014). Specifically, Vamvaka et al. (2014) suggested possible reheating (re-burial) in the Bavarian Forest near the Apatite-BaF sample location during the Jurassic or Lower Cretaceous followed by exhumation in the Upper Cretaceous. To
test this, we set model constraints for scenario 2 such that Jurassic and Cretaceous reburial is permitted but not required (Fig. 6b, d), with the additional limitation that temperatures in the Upper Cretaceous cannot exceed 120 °C (based on apatite fission track data by Vamvaka et al. (2014)). Moreover, we repeated both inversions with $C_{aw}$ calculated from parent nuclide maps with different resolutions for sensitivity testing (cf., Section 3.5.2). The inversions for Ap-BaF-P1 resulted in a large number



of acceptable time-temperature paths for both the cooling-only scenario and the reburial-and-exhumation scenario. However,
good paths were only resolved in the latter and when using $C_{aw}$ calculated from a high-resolution parent nuclide map (Fig. 6d).

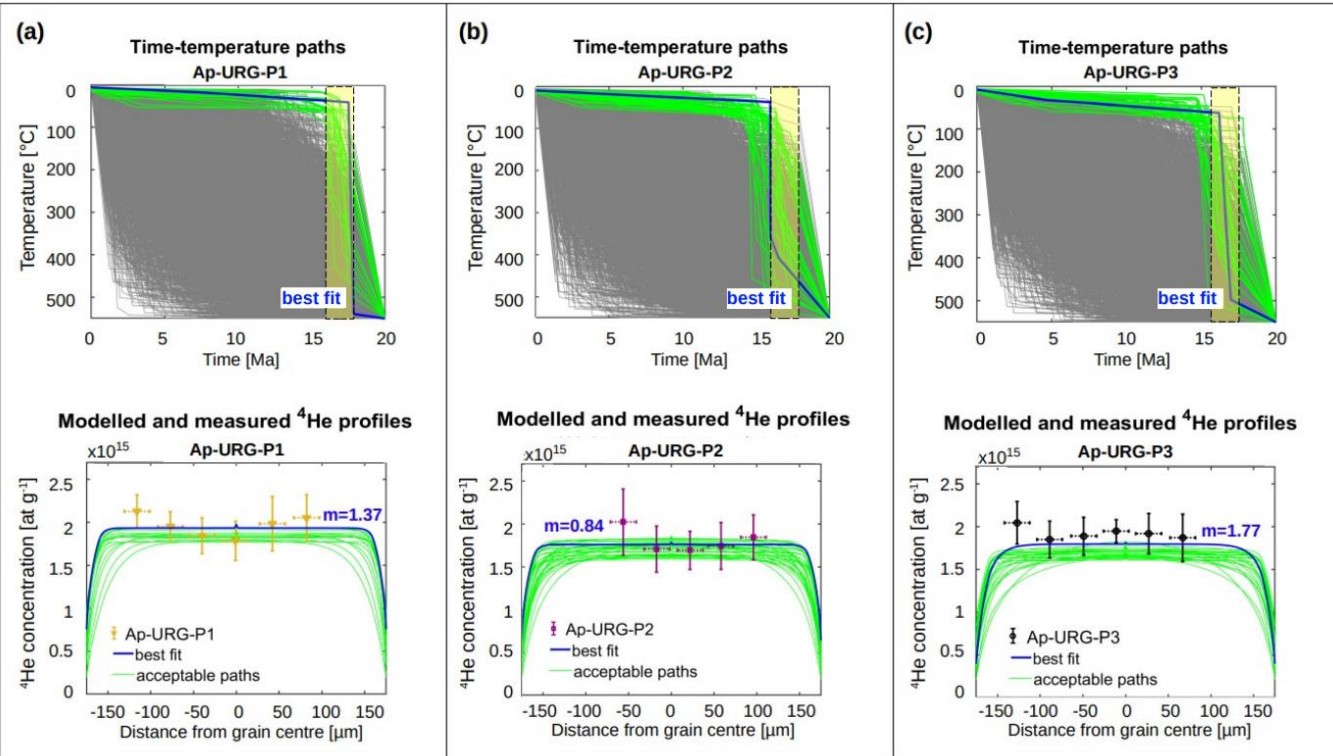

**Figure 5:** Cooling history reconstruction of grain Apatite-URG. The time-temperature (t-T) paths were retrieved by inverting the [4]He profile measurements (upper panels). Based on the acceptable t-T paths, the [4]He profiles were forward-modelled, assuming a homogeneous parent nuclide distribution (lower panels). Acceptable paths (in green) represent a GOF >5%. t-T paths and corresponding [4]He profiles with the lowest misfit m (Section 2.5, Eq.2) are highlighted in blue. The crystallisation date (apatite U-Pb date± 1 standard deviation) of Apatite-URG as determined by Binder et al. (2023) is indicated by a yellow bar in the upper panels.

### 3.5.2 Sensitivity of the thermal models to parent nuclide map resolution

We repeated the inversion for Ap-BaF-P1 for the cooling-only (scenario 1) and for the reburial-and-exhumation scenario (scenario 2) twice to test the sensitivity of the inversion results to the parent nuclide map resolution. The first inversion used $C_{aw}$ calculated from the initial 24x24-µm resolution parent nuclide map, while the second inversion utilised higher-resolution
$C_{aw}$ derived from the interpolated 5x5-µm resolution parent nuclide maps. As mentioned in Section 3.5.1, all four inversions produced acceptable paths. Notably, the misfit between the measured [4]He profile and the forward-modelled [4]He profile based on the best-fitting time-temperature path is lower for the models using 5x5-µm resolution $C_{aw}$ (Fig. 6c, d) than for the models using 24x24-µm resolution $C_{aw}$ (Fig. 6a, b) across both scenario 1 and scenario 2. Further, for the models using 24x24-µm resolution $C_{aw}$ (Fig. 6a, b), the best-fit paths retrieved in scenario 1 and scenario 2 are very similar with misfits of m=2.42
(scenario 1, Fig. 6a) and m=2.45 (scenario 2, Fig. 6b). In contrast, when using the high-





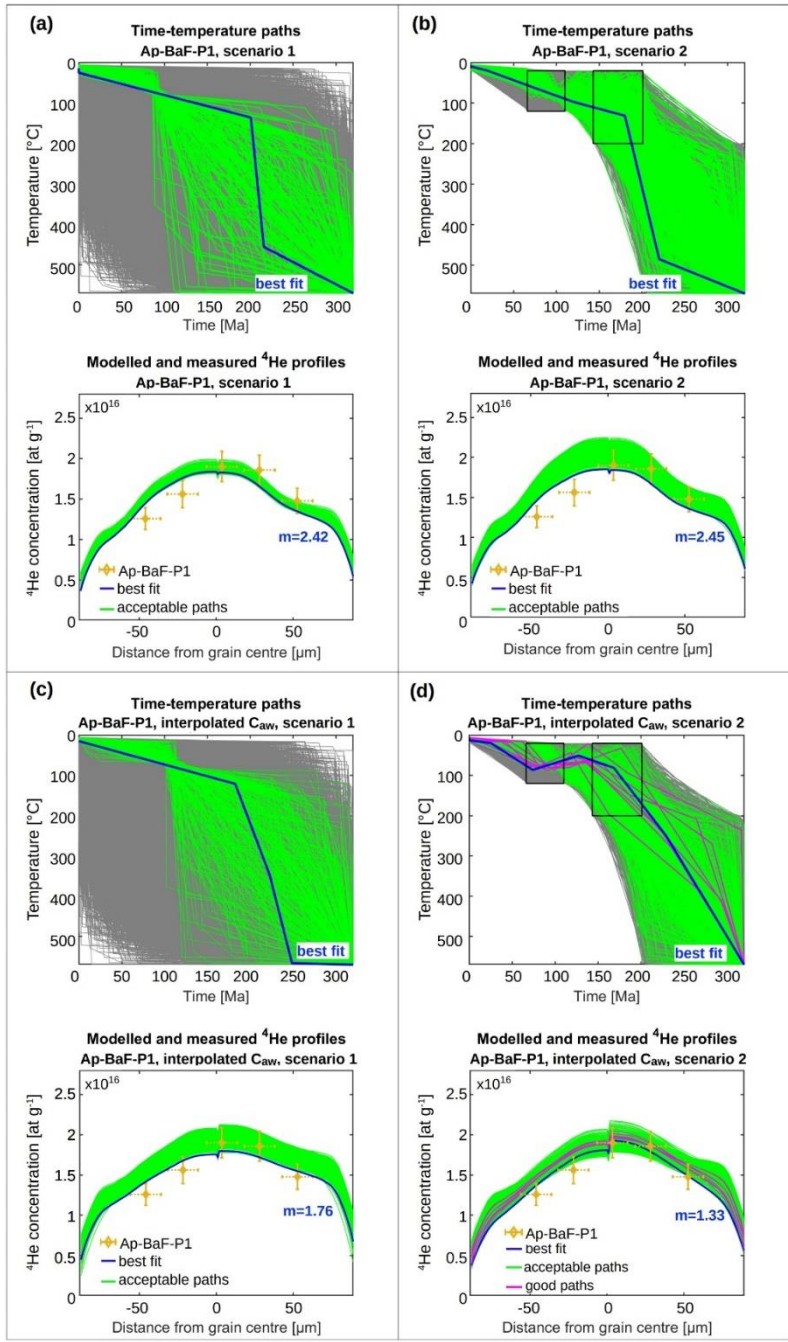

**Figure 6:** Cooling history reconstruction of grain Apatite-BaF testing a cooling-only scenario (scenario 1; a, c) and a reburial-and-exhumation scenario (scenario 2; b, d). The time-temperature (t-T) paths were retrieved by inverting the $^4$He profile measurements and using alpha-stopping-distance weighted parent nuclide concentrations ($C_{aw}$) calculated based on the original 24x24-µm resolution measurements (a,b) and on the interpolated 5x5-µm resolution parent nuclide distributions (c,d). The different resolutions were used to assess the effect of parent nuclide map resolution on thermal modelling. Based on the acceptable t-T paths, the $^4$He profiles were forward-modelled, assuming a heterogeneous parent nuclide distribution. The forward models combine two core-rim profiles, leading to a small jump in the modelled $^4$He concentration in the centre of the grain. Acceptable paths (in green) represent a GOF >5% and good paths (in magenta) represent a GOF >50%. t-T paths and corresponding $^4$He profiles with the lowest misfit m (Section 2.5, Eq.2) are highlighted in blue. The black boxes indicate t-T constraints.   17




resolution $C_{aw}$, the misfit for the best-fit path in scenario 2 (with reheating, m=1.33, Fig. 6d) is distinctly lower than in scenario 1 (cooling only, m=1.76, Fig.6c). Additionally, using high-resolution $C_{aw}$, the misfits for the best-fit paths are in the same range as for the homogeneous Apatite-URG.

**3.6 Summary of the main results**

The preceding paragraphs present the results of in situ $^4$He profile measurements and parent nuclide mapping performed on two apatites from samples in South Germany (Apatite-URG and Apatite-BaF) as well as one apatite grain from the McClure Mountain Syenite standard (Apatite-McClure). We attained $^4$He profiles with individual spot measurement uncertainties of less than 10% for Apatite-BaF (ablation spot diameter 20 μm) and less than 15% for Apatite-URG (ablation spot diameter 30

μm). The measurement uncertainties for Apatite-McClure (ablation spot diameter 10 μm) exceed 40%. Apatite-URG with a homogeneous parent nuclide distribution shows a redundancy between the three measured in situ $^4$He profiles and in situ (U-Th-Sm)/He dates that are generally consistent within measurement uncertainty and overlap with the independently determined apatite U-Pb date of this sample (Binder et al., 2023). Thermal modelling for all $^4$He profiles suggests that Apatite-URG underwent rapid cooling between 15 and 20 Ma.

In contrast, Apatite-BaF with a heterogeneous parent nuclide distribution displays a strong variation in in situ AHe dates from the core (younger) to the rim (older), with the youngest in situ dates corresponding to the areas of highest eU. Only one profile, Ap-BaF-P1, could be inverted to yield acceptable cooling paths. We tested a cooling-only scenario against a scenario of potential Jurassic or Lower Cretaceous reburial followed by Upper Cretaceous cooling as proposed by Vamvaka et al. (2014) for areas near Apatite-BaF's sample location. While the $^4$He profile inversion for both scenarios yielded acceptable time-

temperature paths, good paths were only achieved for the reburial-and-exhumation case, suggesting this to be the more fitting thermal history. Sensitivity testing with $C_{aw}$ calculated from different resolution parent nuclide maps indicates that inverse and forward models using $C_{aw}$ calculated from high-resolution parent nuclide maps produce better results, i.e., a lower misfit between modelled and measured $^4$He profiles, than models using $C_{aw}$ from lower-resolution parent nuclide maps. Lastly, although Apatite-McClure is overall homogeneous in parent nuclides, the high individual measurement uncertainties for $^4$He

do not allow for evaluating its $^4$He profiles.

We make the following general observations that will be further discussed below. (1) There is a strong relation between $^4$He measurement uncertainty and ablation spot size (volume), which needs to be selected to be large enough to reduce analytical uncertainty and small enough to increase spatial resolution. (2) In situ measured $^4$He concentrations and corresponding in situ dates vary with the spot location in the grain and with eU. (3) In situ $^4$He profiles can be inverted for cooling histories of

homogeneous and heterogeneous grains.





# 4 Discussion

## 4.1 Grain size

The direct measurement of (in situ) [4]He profiles requires comparatively large grains, at least 145 µm in diameter in our case. There are two main controls on the minimum analysable grain size: the minimum number of spots needed for a reliable [4]He concentration profile and the minimum ablation spot diameter to reach the required ablation volume (Section 4.3). Regarding the former, our data suggest that at least four evenly spaced measurements (3–5 µm distance from rim to rim of the ablation spot) along a c-axis perpendicular half-profile (core to rim) or six along a rim-to-rim profile are necessary for a reliable [4]He concentration profile. With respect to the latter, we determined, for our laboratory set-up in Tuebingen, an ablation spot diameter of 20 µm as ideal for apatite (Section 4.2). Taken together, for a full profile of six spots with a spot size of 20 µm, a spot spacing of 5 µm and a zero distance between the edge of the outermost ablation spot and the grain rim, the minimum grain diameter is 145 µm. Grains with a low [4]He content (<2.1E15 at g$^{-1}$ in this study), requiring larger ablation spots, can only be analysed if a medium sand-sized fraction is available. This requirement limits the applicability of the single-grain in situ approach for thermal history modelling, especially for small apatites with low parent nuclide concentrations. In such cases where the grain size is small or the required spot size is large (or both), single in situ spots in several grains would have to be used (e.g., Glotzbach and Ehlers, 2024).

## 4.2 Laser ablation spot diameter and pit depth

The choice of ablation spot diameter and pit depth is a compromise between the accuracy of the [4]He concentration profile, which benefits from a smaller spot size and shallower pit, and the analytical uncertainty, which increases with decreasing ablated volume, i.e., decreasing amount of [4]He measured. The lower limit of the ablated volume depends on the [4]He concentration within the grain and specifics of the analysing laboratory concerning [4]He blank levels and criteria for allowable analytical uncertainty. In our case, [4]He measurements should exceed three times the blank level (SD <5%).

Another trade-off exists between smaller-diameter and deeper ablation pits and larger-diameter and shallower ablation pits. The uncertainty introduced by pit volume measurements is one of the limiting factors for the minimum ablation spot size. We determined pit volumes via confocal laser scanning microscopy, which is constrained by the maximum resolvable pit depth at small pit diameter-to-depth ratios. The difficulty with mapping the topography of increasingly narrow and deep pits is illustrated by the progressively higher standard deviations from the mean pit volume in our measurements at lower diameter-to-depth ratios (Table 2). Pickering et al. (2020) found the same type of limitations when using optical interferometry, which demonstrates the need for further development in determining pit volumes. An additional constraint on spot diameter vs. pit depth is a potential parent nuclide zonation. While a small-diameter but deep ablation pit reduces lateral averaging of the helium concentration, it exacerbates the effects of potential 'downhole' parent nuclide zonation and inclusions.

For this study, which includes 98 individual measurements with ablation spot sizes of 10 to 30 µm and corresponding average depths of 7.9 to 9.7 µm (Table 2, B1), a pit diameter of at least 20 µm and depth <8 µm was optimal. Likewise, Pickering et



al. (2020) used 20 µm pit diameters with depths <10 µm for their in situ AHe analysis. For zircons, Danišík et al. (2017) achieved reliable $^4$He measurements for square spots with diameters <10 µm and pit depths of ~2 µm. However, due to the

above factors, we recommend that users conduct test measurements with different ablation pit geometries to determine what suits each sample best before measuring $^4$He concentration profiles.

### 4.3 Laser ablation spot locations in the grain

The placement of $^4$He ablation spots to measure an accurate in situ $^4$He concentration profile for thermal history reconstruction mainly depends on two aspects: the distance to inclusions and fractures, and the distance of the outermost individual spots to

the grain rim. Concerning the former, the distance to inclusions is critical, because mineral inclusions with a potentially many times higher parent nuclide concentration compared to the host crystal may implant foreign helium and lead to excess $^4$He, not directly related to the cooling history, in the surrounding grain (e.g., Vermeesch et al., 2007). Furthermore, fractures or voids can trap $^4$He and locally affect the $^4$He diffusion kinetics (e.g., Zeitler et al., 2017). As these phenomena complicate cooling history reconstructions, their periphery should be avoided. When selecting $^4$He ablation spots, a minimum distance of 20 µm

from inclusions or fractures (for average alpha-stopping distances, e.g., Pickering et al., 2020) should be maintained. Still, if possible, grains with these features should not be analysed. We discuss the effect of grain heterogeneities further in Section 4.5.

More crucial for $^4$He profile measurements is the distance of a $^4$He ablation spot to the grain rim, provided an adequate grain is selected. Close to the grain rim, $^4$He measurements will average concentrations across a steep gradient (depending on the

spot size) due to alpha-ejection at the grain boundary (e.g., Farley et al. 1996; Farley, 2002). This leads to a decreased accuracy of the measurements near the rim. To avoid grain rim effects and to account for the full range of alpha-stopping distances, an ablation spot would need to be at least 40 µm away from the grain boundary (distance from the ablation spot centre to the grain rim). However, this poses a problem since the shape of the helium profile near the grain rim is diagnostic for differentiation between slow and fast cooling. Ultimately, the difference between a flat (fast-cooled) $^4$He profile and a rounded (slow-cooled)

$^4$He profile is best observed at the grain rim (Shuster and Farley, 2004). Not measuring $^4$He within 40 µm of the grain rim would thus exclude characteristic information. In this exploratory study, we measured $^4$He closer than 40 µm to the grain rim (Fig. 2) but did not calculate alpha-stopping distance weighted parent nuclide concentrations ($C_{aw}$) for those spots or use them for the $^4$He profile inversion. Nevertheless, we included those measurements for comparisons between the measured and forward-modelled $^4$He profiles. Further studies are needed to determine best practices concerning the $^4$He spot placement and

measurements near the grain rim.

Furthermore, our results for Apatite-URG (Fig. 2a) suggest that in homogeneous grains, the placement of the profile closer to the grain tips or middle does not influence the in situ $^4$He profile's shape. Information gathered from multiple profiles in such cases is expected to be redundant, as demonstrated in all Ap-URG profiles (Fig. 2a) and in three of four Ap-BaF profiles that are indistinguishable within measurement error (Fig. 2b). Hence, for homogeneous or concentrically zoned grains, it may





suffice to measure a half-profile. However, we still recommend analysing 2–3 rim-to-rim profiles because the likelihood of detecting anomalies in parent nuclide and $^4$He distribution, e.g., due to inclusions, is higher.

### 4.4 Spatial variation of in situ dates in a grain

The two analysed apatites Apatite-URG and Apatite-BaF both display a variation of in situ AHe dates from core to rim, with a trend of older dates towards the grain rim and younger dates towards the grain centre (Fig. 3h, l). In Apatite-URG, this

tendency of the measured in situ dates towards older dates at the grain rim (Fig. 3h) is not pronounced, as the dates are generally consistent within measurement uncertainty. Further, the weighted mean AHe date is within the apatite U-Pb date determined by Binder et al. (2023) for this sample (cf. Section 3.4). In the heterogeneous Apatite-BaF, the pattern of older dates at the grain rim (Fig. 3l) is distinct, with the dates at the rim being up to twice as old as the dates in the centre.

In both cases, the observed date distribution within the grains is counterintuitive. While the apparent overlap of AHe and

apatite U-Pb dates in Apatite-URG can be anticipated for a volcanic sample (Apatite-URG is from a foiditic tuff; Table 1), where the time for $^4$He diffusion is limited by short residence times in the subvolcanic system, the trend towards older dates at the grain rim is unexpected. In theory, the oldest in situ date in a homogeneous crystal such as Apatite-URG should be in the centre due to (uniform Arrhenius-type) diffusion leading to a relative depletion of $^4$He in the rims compared to the core (Glotzbach and Ehlers, 2024). A pattern of younger dates nearer to the rim would also be logical for a heterogeneous grain

like Apatite-BaF where the parent nuclides are relatively enriched in the core compared to the rim (Fig. 3 i-l). Here, the rims should be depleted in $^4$He compared to the core, even when considering radiation damage effects (e.g., Shuster et al. 2006) and hence yield younger in situ dates. From our data, we cannot decipher the reason for the observed inverted in situ date pattern. It is unclear whether the older dates near the grain rims in both grains are outlier measurements or if they result from undetected local grain heterogeneities. Possible reasons for the old dates include a locally high alpha-particle production in the polished-

away portion of the grain or from deeper in the unanalysed remaining grain fraction. It could also be due to a higher, local $^4$He retentivity in the crystal lattice. In our modelling approach, we can only account for the redistribution of $^4$He from the radioactive decay event via $C_{aw}$ calculation. Any other processes that could locally deplete or enrich $^4$He and lead to older in situ dates (e.g., lattice defects trapping $^4$He) and alter the diffusive behaviour are not considered. Imaging techniques such as Raman spectroscopy would be necessary for further investigation and refinement.

### 4.5 Parent nuclide heterogeneity

Previous studies have evaluated the influence of parent nuclide zonation on $^4$He profile thermal modelling in the context of whole grain $^4$He/$^3$He analyses. They demonstrated that undetected and unquantified zonation of parent nuclides can result in retrieving incorrect cooling histories since parent nuclide heterogeneities do not always visibly manifest in the shape of the measured $^4$He profile but still affect the $^4$He concentration and distribution in the grain (e.g., Shuster and Farley, 2004; Farley

et al., 2010). Hence, mapping the parent nuclide distribution of exposed internal grain surfaces is crucial in assessing the extent of parent nuclide heterogeneity influencing the $^4$He distribution (e.g., Farley et al., 2011; Danišík et al., 2017).





In this study, Apatite-BaF exemplifies a case where the impact of parent nuclide zonation is not apparent from the measured [4]He profiles' shapes. The profiles Ap-BaF-P1, Ap-BaF-P2 and Ap-BaF-P4 (Fig. 2b) display an inconspicuous shape with a smooth decrease in [4]He concentration from the grain centre to the rim, typical for slowly cooled grains (Shuster and Farley,
2004), save for a slight skewing of the maximum concentration off-centre for Ap-BaF-P2 and Ap-BaF-P4. Even so, the comparison of measured and modelled [4]He profiles (Figs. 6a, b and 7) indicates that the [4]He gradient measured near the grain rim is not achievable solely by finding fitting time-temperature paths. The apparent discrepancy between measured and modelled [4]He profiles near the grain rim, more so in the left side than the right (Fig. 6a, b, Fig.7), suggests a significant influence of parent nuclide heterogeneity (Figs. 2 and 3 i–l) and associated variations in the [4]He production and diffusion in
the crystal (e.g., Farley et al., 2010). This underlines that determining the parent nuclide distribution is a necessary step in interpreting in situ [4]He concentration profiles (e.g., Farley et al., 2011; Danišík et al., 2017; Fox et al., 2017).

**4.6 Influence of parent nuclide map resolution on thermal modelling**

Mapping the parent nuclide concentration on the exposed internal grain surface via LA-ICP-MS allows treating the in situ [4]He concentration as a function of the surrounding parent nuclide distribution to achieve more accurate [4]He profile-parent nuclide
relationships for heterogeneous grains (e.g., Farley et al., 2010; Danišík et al., 2017). By using the alpha-stopping distance weighted parent nuclide concentration $C_{aw}$ derived from such parent nuclide maps for [4]He profile thermal modelling, we can also account for the redistribution of [4]He from high-energy alpha decay (Section 2.4).

To illustrate the effect of parent nuclide heterogeneity on in situ [4]He profiles and as a first assessment of the thermal models' sensitivity to the parent nuclide map resolution, we compare forward-modelled [4]He profiles based on the same time-
temperature path but assuming different parent nuclide distributions in Figure 7. As an example for a homogeneous grain, we compare the forward model results for Apatite-URG using a uniform parent nuclide distribution calculated as an average of all parent nuclide measurements (red curve, Fig. 7a), and using $C_{aw}$ calculated from the 24x24-µm resolution parent nuclide map (blue curve, Fig. 7a). For the heterogeneous Apatite-BaF, we conducted the same tests and added a forward model using $C_{aw}$ calculated from the higher-resolution, 5x5 µm, interpolated parent nuclide map (black line, Fig. 7b). We arbitrarily chose
the best-fit time-temperature path retrieved by the respective inverse models in Figure 5a (Apatite-URG) and Figure 6c (Apatite-BaF) as a fixed input cooling history for the forward model tests. Figure 7a shows that for the mostly homogeneous Apatite-URG, the forward-modelled [4]He concentration profile using $C_{aw}$ (blue curve, misfit=1.30; Fig. 7a) does not differ much from the forward-modelled [4]He profile based on an averaged, uniform parent nuclide distribution (red curve, misfit=1.37; Fig. 7a). The slight concave-up pattern of the measured [4]He profile (yellow data points, Fig. 7a), however, can solely be
modelled with $C_{aw}$. In contrast, for the asymmetrically-zoned grain Apatite-BaF the shapes of the forward-modelled [4]He profiles differ significantly for the different parent nuclide distributions (Fig. 7b). Here, the forward-modelled [4]He profile based on the averaged uniform parent nuclide distribution (red curve, m=4.31; Fig. 7b) is too flat and does not capture the curvature of the measured [4]He profile (yellow data points, Fig. 7b) compared to using $C_{aw}$, as calculated from the measured 24x24-µm resolution (blue curve, m=4.15; Fig. 7b) and from the interpolated 5x5-µm resolution parent nuclide maps (black



curve, m=1.76; Fig. 7b). Further, while the forward-modelled $^4$He profile using the original 24x24-µm resolution-based $C_{aw}$

captures the measured $^4$He profile's shape in the right half of the grain, it overestimates the $^4$He concentration in the left side

of the grain. This is consistent with observations from the thermal modelling results shown in Figures 6a and 6b. The best

results were achieved with the 5x5-µm resolution-based $C_{aw}$ (Figs. 6c, d and Fig. 7).

In summary, while for homogeneous grains the difference in modelling results assuming a uniform parent nuclide distribution

or $C_{aw}$ is small, the parent nuclide distribution has a significant influence on the $^4$He profile in heterogeneous grains. Further,

it appears that models with $C_{aw}$ from higher-resolution maps yield better results than models with $C_{aw}$ from lower-resolution

maps. However, evidence from one grain is limited and only a first step towards a systematic investigation into the optimal

resolution for parent nuclide measurement and interpolation. Moreover, parent nuclide concentration interpolation and

assumptions made in the calculation of $C_{aw}$ (Section 2.4) introduce uncertainties, whose influence needs to be tested in future

studies. To calculate $C_{aw}$, we assume that the grain's parent nuclide distributions are mirror-symmetric about the exposed

internal surface due to half of the grain being lost during the grinding and polishing steps of sample preparation. Second, we

assume that our determined ablation time-depth relationship holds (Section 2.3). Further uncertainty is introduced when

localising the $^4$He ablation spot centres on the LA-ICP-MS element maps, which is particularly critical for spots near the grain

rim, where the interpolated grain boundary of the parent nuclide map does not always accurately capture the real grain

boundary. Further studies are also required to test the optimal resolution and the necessity of element maps of the entire grain.

Regarding the latter, it might suffice to map the 40-µm proximity of the $^4$He profile, covering the full alpha-stopping distance

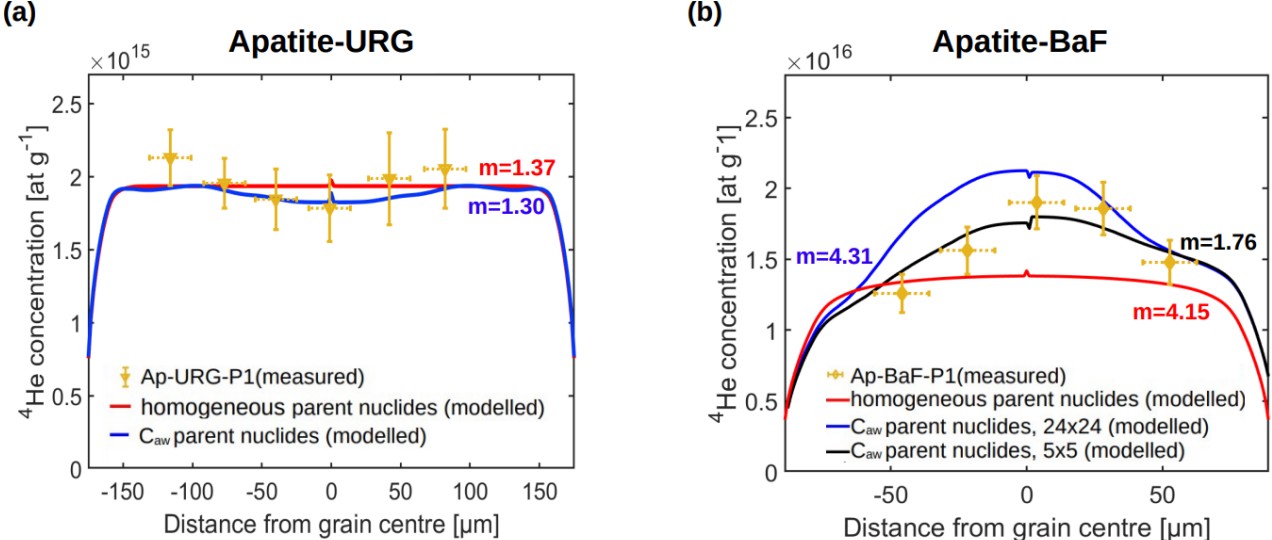

**Figure 7**: Influence of parent nuclide zonation on forward-modelled $^4$He profiles. The profiles in (a) were forward-modelled based on the best-fit-path of profile Ap-URG-P1 (Fig. 5a), and the profiles in (b) were forward-modelled based on the best-fit path of Ap-BaF-P1 that resulted from the thermal history inversion in Figure 6c. The red curves are forward-modelled $^4$He profiles assuming a grain-averaged homogeneous parent nuclide concentration, the blue curves are forward-modelled $^4$He profiles using the alpha-stopping distance weighted parent nuclide concentration $C_{aw}$ calculated from the uninterpolated parent nuclide maps in both grains, and the black curve in (b) is the forward-modelled $^4$He profile for Ap-BaF-P1 with $C_{aw}$ based on the interpolated, higher resolution 5x5 µm element maps. m denotes the misfit between modelled and measured $^4$He profiles (Section 2.5, Eq. 2).





range. This would account for heterogeneities more efficiently, although information on potential element zonations of the entire grain surface would then not be available. This approach could be augmented using other imaging techniques, such as cathodoluminescence, and Raman spectroscopy, to detect factors potentially affecting the $^4$He diffusivity (e.g., Ault and Flowers, 2012; Danišík et al., 2017).

### 4.7 Cooling history reconstruction from single grains

We demonstrated through analyses of a homogenous apatite (Apatite-URG) and a heterogeneous apatite (Apatite-BaF) that the combination of situ $^4$He measurements and $C_{aw}$ calculated from element maps can be inverted for cooling histories of single grains. The example of Apatite-URG shows that the $^4$He profile of a fast-cooled homogeneous grain as young as 16 Ma can be retrieved from six in situ spot measurements and its cooling history can be accurately determined based thereon (Fig. 5). The example of Apatite-BaF shows that $^4$He profiles of heterogeneous grains are more challenging to invert. Here, only one out of four $^4$He profiles (Ap-BaF-P1, Fig. 2) could be successfully inverted for potential cooling histories. Even so, the inversion of Ap-BaF-P1 with high-resolution $C_{aw}$ (Fig. 6d), resulted in a misfit between the forward-modelled and measured $^4$He profiles comparable to results from the homogeneous Ap-URG. This suggests a potential for routine analysis of heterogeneous grains with the in situ method, pending further refinement.

Additional challenges in inverting for time-temperature paths of heterogeneous grains arise from the current model implementation, which predicts c-axis-symmetric $^4$He profiles based on in situ measurements (Glotzbach and Ehlers, 2024). Therefore, we could only successfully invert the least asymmetric half-profile section in Apatite-BaF (Ap-BaF-P1, Fig. 6). For the forward models of Ap-BaF-P1 (Fig. 6, Fig. 7b), we adjusted the model to merge two core-rim profiles into a fully asymmetric rim-rim profile. We did not implement this approach in the inverse model; however, this could be the subject of future studies. Additionally, further studies are needed to examine the effects of local changes in diffusivity mentioned in Section 4.4, such as the impact of radiation damage and whether this inhibits the modelling of heterogeneous grains.

### 4.8 Comparison with other single-grain thermal history reconstruction approaches

Our in situ $^4$He profile approach is conceptually similar to the whole-grain $^4$He/$^3$He method by Shuster and Farley (2004) and the in situ element-maps to 1D-profile method by Danišík et al. (2017). A key difference between the $^4$He/$^3$He approach and the in situ methods is that the in situ approaches enable direct measurements of $^4$He profiles. In contrast, the $^4$He/$^3$He method requires proton irradiation of the samples to create a synthetic uniform $^3$He distribution before helium measurement by step-wise degassing (cf., Shuster and Farley, 2004). This difference is crucial because the need for proton irradiation currently limits the accessibility of $^4$He/$^3$He analyses (e.g., Colleps et al. 2024).

Danišík et al. (2017), who pioneered the concept of cooling history inversion from an in situ measured $^4$He profile in zircon, illustrated that another advantage of in situ mapping of $^4$He and parent nuclides compared to the whole-grain $^4$He/$^3$He measurement lies in the ability to analyse the spatial relationship between parent and daughter isotopes, as failing to account for the effect of grain heterogeneities on $^4$He profiles can lead to inaccurate thermal models (Danišík et al. 2017).



Our approach differs from the protocol of Danišík et al. (2017) in that we do not perform [4]He and parent nuclide concentration
mapping across the entire grain surface and convert those maps into 1D equivalent-sphere profiles. Instead, we directly obtain
the [4]He profiles from spot measurements along c-axis-perpendicular transects through the grain and combine them with parent
nuclide mapping. This method requires fewer individual [4]He analyses, improving efficiency. Furthermore, by integrating the
[4]He profiles with $C_{aw}$ from the element maps recorded at different "downhole" ablation depths, we can better understand the
three-dimensional redistribution of [4]He and account for long alpha-stopping distances.
Even though further studies are needed to test the reliability of the in situ profile method, for example, by comparing results
from different grains of the same sample, we think it provides a useful additional tool for cooling history reconstruction,
especially for samples where grains of variable kinetics (i.e., grain sizes or eU) are not available to constrain possible time-
temperature paths (for whole grain (U-Th-Sm)/He analyses) and where intracrystalline heterogeneities are prevalent.

## 5 Conclusion

In this exploratory study, we tested a new approach to obtain [4]He profiles in apatite from in situ measurements and model the
cooling histories of single apatite grains. We examined the limitations regarding the location, size and number of ablation
spots, as well as the grain size needed to measure an interpretable in situ [4]He profile for our laboratory set-up in Tuebingen.
Further, we introduced $C_{aw}$, an alpha-stopping distance weighted parent nuclide concentration at each ablation site, calculated
from 2D trace element maps, to allow for thermal modelling from in situ [4]He measurements. We demonstrated the feasibility
of our new approach on two natural apatite grains (one homogeneous, one zoned) from South Germany. From these results,
we conclude the following:

1.    The measurement of reliable [4]He profiles using the in situ (U-Th-Sm)/He approach is limited by minimum
      requirements on grain size and ablated volume. For our laboratory set-up in Tuebingen, we find apatites larger
      than 145 µm with [4]He concentrations greater than 1E16 at g[-1] are most suitable to achieve satisfactory results.
555          These dimensions may vary among different laboratories.

2.    Our data indicate that obtaining a [4]He concentration profile requires at least four measurements from the grain
      core to the rim or six from rim to rim.

3.    Using LA-ICP-MS parent nuclide mapping helps detect intracrystalline heterogeneities. The calculation of $C_{aw}$
      is crucial in analysing heterogeneous grains, yet may be unnecessary in homogeneous grains where the benefit
560          of $C_{aw}$ calculations compared to using an averaged homogeneous parent nuclide concentration is marginal. This
      is important since parent nuclide mapping, inversion for 2D maps, and $C_{aw}$ calculation can be time-consuming.
      To improve efficiency, one possibility is to map the 40 µm perimeter surrounding the [4]He spots for parent
      nuclides instead of the entire grain surface. This approach would suffice for the calculation of $C_{aw}$. However, it
      limits the information available on grain zonation patterns and crystal lattice heterogeneities, which could be



vital for interpreting asymmetric [4]He profiles. Therefore, the trade-off between efficiency and potential information loss needs to be systematically tested.

4.    Cooling histories can be inverted from in situ [4]He profiles and parent nuclide maps. While the method is readily applicable to homogeneous grains, the inversion of asymmetric [4]He profiles (heterogeneous grains) would benefit from further studies.

**Code and data availability**

The code to calculate $C_{aw}$, along with a test file, and supplementary data including all grain photomicrographs, He measurement details, and all U, Th, and Sm measurements, can be found here: https://doi.org/10.5281/zenodo.15856623

**Author contributions**

AKM: data curation, formal analysis, investigation, methodology, software, visualisation, writing – original draft; CG: conceptualisation, methodology, funding acquisition, resources, software, supervision, validation, writing – review & editing; SF: supervision, writing – review & editing.

**Competing interests**

The authors declare that they have no conflict of interest.

**Acknowledgements**

We thank Dominic Raisch from the Petrology and Mineral Resources Research Group at the University of Tübingen for SEM imaging. This study was supported by a grant from the Bundesgesellschaft für Endlagerung to Christoph Glotzbach (BGE – STAFuE-21-12-Klei), and funding for large equipment from the DFG (INST 37/1041-1 FUGG).

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



## Appendix A: Additional depth-resolved interpolated parent nuclide maps of Apatite-URG and Apatite-McClure

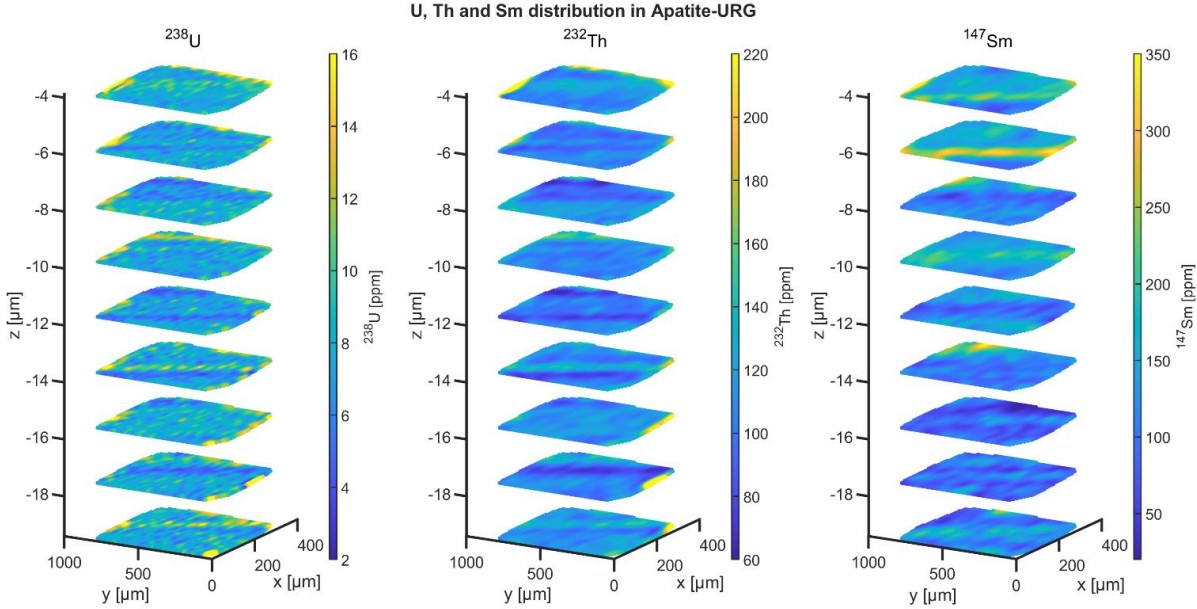

715

**Figure A1:** Interpolated parent nuclide distribution maps (10x10 µm horizontal resolution) of Apatite-URG. Vertically, the parent nuclide concentrations were recorded approximately every 2 µm for a 20 µm deep section in the grain. Parent nuclide maps were interpolated with a smoothness constraint of $\lambda=0.1$ for the $^{238}U$ and $^{232}Th$ and $\lambda=0.01$ for the $^{147}Sm$ maps.

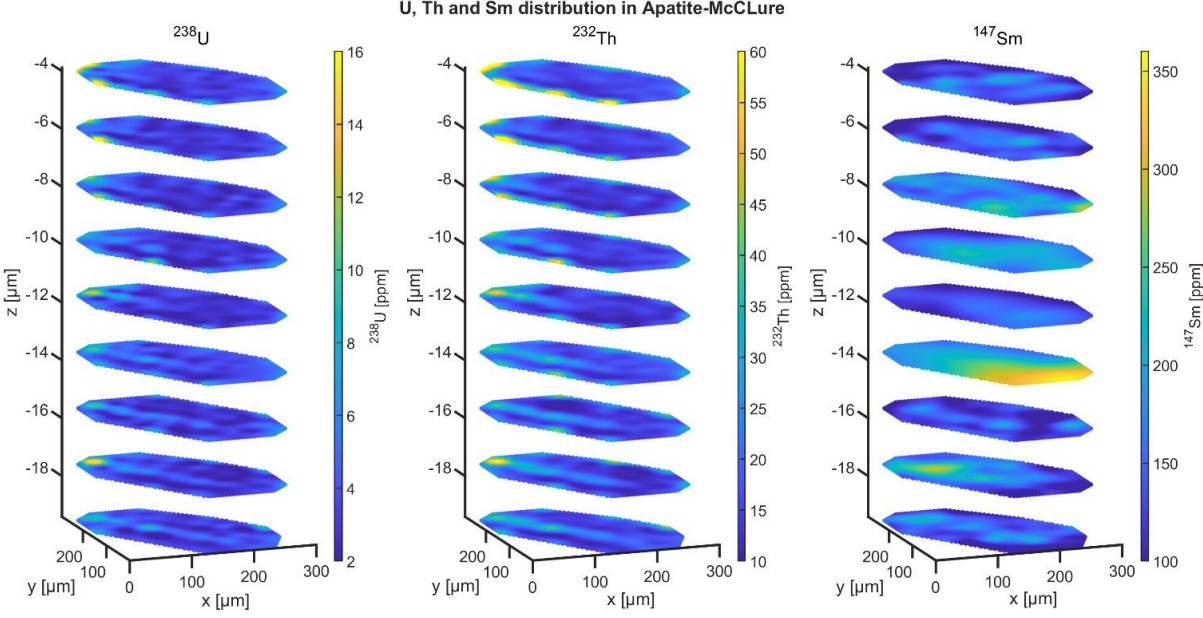

720

**Figure A2:** Interpolated parent nuclide distribution maps (5x5 µm horizontal resolution) of Apatite-McClure. Vertically, the parent nuclide concentrations were recorded approximately every 2 µm for a 20 µm deep section in the grain. Parent nuclide maps were interpolated with a smoothness constraint of $\lambda=0.3$ for the $^{238}U$ and $^{232}Th$ and $\lambda=0.01$ for the $^{147}Sm$ maps.



## Appendix B: Apatite in situ $^4$He measurements

**Table B1: Apatite-McClure $^4$He data and alpha-stopping distance weighted parent nuclides concentrations ($C_{aw}$)**

| Spot | Pit volume [μm$^3$] | Pit depth [μm] | $^4$He [at g$^{-1}$] | $^4$He SD [at g$^{-1}$] | $^{238}$U $C_{aw}$ ± 1SD [ppm][a] | $^{232}$Th $C_{aw}$ ± 1SD [ppm][a] | $^{147}$Sm $C_{aw}$ ± 1SD [ppm][a] | Distance to grain boundary [μm][b] |
|---|---|---|---|---|---|---|---|---|
| Ap-McClure_1 | 289 | 9.3 | 8.58E+15 | 5.20E+15 | - | - | - | 56 |
| Ap-McClure_2 | 332 | 9.5 | 5.27E+15 | 4.52E+15 | - | - | - | 60 |
| Ap-McClure_3 | 310 | 9.7 | 6.10E+15 | 2.39E+15 | 3.8 ± 0.4 | 16.9 ± 1.5 | 179 ± 27 | 63 |
| Ap-McClure_4 | 301 | 9.4 | 4.91E+15 | 3.88E+15 | 3.5 ± 0.3 | 16.5 ± 1.5 | 162 ± 20 | 66 |
| Ap-McClure_5 | 307 | 9.3 | 5.88E+15 | 2.90E+15 | 3.4 ± 0.4 | 16.1 ± 2.0 | 162 ± 12 | 68 |
| Ap-McClure_6 | 231 | 9.4 | 4.24E+15 | 2.81E+15 | 3.0 ± 0.3 | 15.1 ± 1.5 | 182 ± 43 | 71 |
| Ap-McClure_7 | 302 | 9.4 | 4.47E+15 | 2.40E+15 | 3.2 ± 0.4 | 15.8 ± 2.5 | 189 ± 28 | 74 |
| Ap-McClure_8 | 305 | 9.3 | 4.83E+15 | 3.14E+15 | 3.3 ± 0.4 | 16.8 ± 2.9 | 164 ± 5 | 74 |
| Ap-McClure_9 | 308 | 10.0 | 8.60E+15 | 2.45E+15 | - | - | - | 45 |
| Ap-McClure_10 | 311 | 10.1 | 7.38E+15 | 3.26E+15 | - | - | - | 49 |
| Ap-McClure_11 | 325 | 9.2 | 5.36E+15 | 2.46E+15 | - | - | - | 52 |
| Ap-McClure_12 | 317 | 10.0 | 8.42E+15 | 4.07E+15 | 5.5 ± 0.8 | 26.8 ± 6.0 | 112 ± 8 | 56 |
| Ap-McClure_13 | 344 | 9.9 | 5.64E+15 | 4.94E+15 | 5.3 ± 0.6 | 26.6 ± 4.9 | 114 ± 23 | 59 |
| Ap-McClure_14 | 359 | 9.7 | 7.59E+15 | 3.95E+15 | 5.1 ± 0.4 | 26.1 ± 3.4 | 116 ± 37 | 63 |
| Ap-McClure_15 | 337 | 9.8 | 4.53E+15 | 3.01E+15 | 4.8 ± 0.5 | 24.9 ± 4.1 | 129 ± 31 | 66 |
| Ap-McClure_16 | 303 | 10.1 | 4.61E+15 | 2.51E+15 | 4.7 ± 0.9 | 25.1 ± 6.4 | 158 ± 13 | 70 |
| Ap-McClure_17 | 330 | 9.7 | 4.25E+15 | 2.13E+15 | 4.6 ± 0.9 | 25.0 ± 6.9 | 153 ± 12 | 73 |
| Ap-McClure_18 | 270 | 9.9 | 6.22E+15 | 2.76E+15 | - | - | - | 9 |
| Ap-McClure_19 | 315 | 9.6 | 2.77E+15 | 5.01E+15 | (3.2 ± 0.3) | (15.4 ± 1.9) | (151 ± 43) | 24 |
| Ap-McClure_20 | 320 | 9.7 | 5.56E+15 | 4.21E+15 | (3.7 ± 0.7) | (19.9 ± 5.1) | (178 ± 55) | 39 |
| Ap-McClure_21 | 343 | 9.5 | 5.63E+15 | 4.39E+15 | 4.7 ± 0.9 | 25.8 ± 6.3 | 168 ± 30 | 54 |
| Ap-McClure_22 | 308 | 9.2 | 3.08E+15 | 4.60E+15 | 3.8 ± 0.6 | 19.4 ± 3.7 | 132 ± 24 | 65 |
| Ap-McClure_23 | 279 | 9.5 | 4.88E+15 | 3.82E+15 | 3.4 ± 0.4 | 16.2 ± 1.9 | 124 ± 13 | 49 |
| Ap-McClure_24 | 322 | 9.2 | 6.35E+15 | 2.24E+15 | (3.4 ± 0.4) | (15.2 ± 1.8) | (128 ± 11) | 35 |





| | | | | | | | | |
|---|---|---|---|---|---|---|---|---|
| Ap-McClure_25 | 304 | 9.3 | 6.37E+15 | 2.94E+15 | (3.5 ± 0.4) | (15.1 ± 1.5) | (131 ± 9) | 19 |
| Ap-McClure_26 | 265 | 9.7 | 6.98E+15 | 2.99E+15 | - | - | - | 43 |
| Ap-McClure_27 | 354 | 9.9 | 5.46E+15 | 2.62E+15 | 5.5 ± 1.0 | 26.4 ± 6.4 | 117 ± 12 | 60 |
| Ap-McClure_28 | 363 | 9.7 | 6.12E+15 | 3.53E+15 | 6.2 ± 0.8 | 30.5 ± 6.3 | 118 ± 5 | 74 |
| Ap-McClure_29 | 219 | 10.1 | 6.93E+15 | 4.19E+15 | - | - | - | 38 |
| Ap-McClure_30 | 321 | 9.9 | 6.47E+15 | 2.79E+15 | - | - | - | 21 |
| Ap-McClure_31 | 323 | 9.6 | 6.77E+15 | 4.14E+15 | - | - | - | 4 |
| Ap-McClure_32 | 301 | 10.2 | 4.34E+15 | 3.68E+15 | (3.2 ± 0.4) | (15.6 ± 2.6) | (140 ± 15) | 36 |
| Ap-McClure_33 | 299 | 9.7 | 5.41E+15 | 3.12E+15 | 3.4 ± 0.4 | 16.5 ± 2.2 | 138 ± 22 | 52 |
| Ap-McClure_34 | 296 | 9.9 | 4.93E+15 | 2.48E+15 | 3.4 ± 0.3 | 16.4 ± 1.7 | 149 ± 31 | 68 |
| Ap-McClure_35 | 330 | 9.4 | 5.25E+15 | 2.77E+15 | 3.5 ± 0.4 | 15.8 ± 1.7 | 167 ± 12 | 50 |
| Ap-McClure_36 | 325 | 9.8 | 6.72E+15 | 2.99E+15 | (3.2 ± 0.4) | (15.1 ± 1.5) | (128 ± 9) | 34 |
| Ap-McClure_37 | 323 | 9.6 | 4.49E+15 | 3.33E+15 | (3.8 ± 0.8) | (19.5 ± 5.9) | (127 ± 7) | 18 |
| Ap-McClure_38 | 327 | 10.0 | 5.03E+15 | 2.83E+15 | - | - | - | 2 |

725    [a] Alpha-stopping distance weighted parent nuclide concentrations ($C_{aw}$; see Section 2.4) were not calculated for spots less than the maximum alpha-stopping distance of 40 µm away from the grain boundary, and for spots that were measured along a c-axis parallel traverse (Section 2.5). Note that locating the [4]He spots on the parent nuclide map is subject to uncertainty, especially for non-straight grain boundaries. Thus, the $C_{aw}$ calculation for spots close to the grain rim needs to be treated with caution. Where the interpolated parent nuclide map adds area to the grain, $C_{aw}$ values are reported in round brackets.

730    [b] c-axis orthogonal distance from the He-measurement spot centre to the nearest grain rim.

For Apatite-McClure in situ dates were not calculated due to the [4]He measurements' high standard deviation (SD).