# Peer review of "Analytical and modelling strategies for thermal histories from in situ (U-Th-Sm)/He data of single apatites"

_EGUsphere, 2025_

## Author Comment (AC1)

**Authors' response to Review 1**

- Reviewer: Olga Yakubovich
- Citation: https://doi.org/10.5194/egusphere-2025-3879-RC1

Note: The reviewer comments are presented in **black bold print** in their original, unabridged wording. The author's response is displayed in blue.

**The submitted manuscript is well written and presents a new methodological approach and algorithm for in situ (U–Th–Sm)/He dating of apatite aimed at reconstructing thermal histories of individual grains. Using apatite samples from southern Germany, the authors demonstrate that the proposed methodology enables the derivation of "He diffusion" profiles. It is shown that profiles corrected for parent nuclide distribution are flat in rapidly cooled grain, whereas in samples with heterogeneous parent nuclide distribution and complex thermochronological history, they exhibit non-flat geometries. It is shown how the obtained analytical data can be mathematically processed to derive information on the thermal history of the grains.**

**Thus, I consider this study to represent a valuable advancement in the development of in situ (U–Th–Sm)/He thermochronology.**

We thank the reviewer for their time and careful review of our manuscript. The reviewer raises the following key concerns in their general comments below: Firstly, they state that for one of the apatites we analysed, we did not succeed in deriving a thermochronological path matching the observations in that grain. Secondly, they suggest that analysing well-characterised standard material would have been preferable to our choice of sample material. Lastly, the reviewer notes that some of our methodological approaches require a more detailed description. We address all comments below in detail and are optimistic that our answers and proposed revisions to the manuscript will alleviate the reviewer's concerns.

We have numbered the following reviewer comments (RC) and author responses (AR).

- **RC 1: My major concern is that the authors did not succeed in deriving a thermochronological path that adequately fits their observations for the BaF apatite. The obtained 4 profiles are reproducible and, as the authors noted, exhibit a counterintuitive geometry: the central part of the grain is significantly younger than the rims.**
- **AR 1:** We thank the reviewer for this comment. As they bring up several points at once, we will address each issue separately.
  - On the lack of success for deriving an adequate thermochronological path:
    Regarding the challenges in deriving an adequate thermochronological path for Apatite-BaF, we acknowledge that we were unable to invert all [4]He profiles of

Apatite-BaF for cooling histories. However, we want to emphasise that for profile Ap-BaF-P1, we did obtain fitting cooling histories (see Fig. 6 in our manuscript). We think a major issue with the non-invertible $^4$He profiles is their asymmetry with respect to the c-axis, which conflicts with our inverse modelling approach that assumes approximately c-axis symmetric $^4$He profiles. We will revise Section 4.7 in the discussion to highlight this limitation in our modelling approach. Furthermore, we suggest running forward models for Ap-BaF-P2 to Ap-BaF-P4 using the inversion results of Ap-BaF-P1 to assess whether the cooling history derived from Ap-BaF-P1 would fit the non-invertible profiles in Apatite-BaF.

- o On the counterintuitive geometry of the $^4$He profiles:
  For the $^4$He profiles of Apatite-BaF, we observe that the $^4$He concentration is highest at or near the grain centre, and the lowest at the grain rims. The profiles show a concave down geometry, as expected for a grain that has experienced diffusive loss of $^4$He. The only exception to this pattern is Ap-BaF-P3, where the centremost spot exhibits the lowest $^4$He concentration (Fig. 2 and Section 3.2, lines 232-237). We did not intend to suggest that the geometry of the $^4$He profiles in Apatite-BaF (except for Ap-BaF-P3) is counterintuitive. We will clarify in Section 3.2 that, overall, the $^4$He profiles for Apatite-BaF are rather inconspicuous.

- o On the central part of the grain being younger than the grain rims:
  The pattern of younger in situ AHe dates in the grain centre of Apatite-BaF compared to the grain rim is counterintuitive and is most pronounced in Ap-BaF-P2 and Ap-BaF-P3. Ap-BaF-P1, which we successfully inverted for cooling histories, does not show this trend. Thus, the overall puzzling AHe date pattern did not preclude the derivation of potential time-temperature paths for Apatite-BaF. We will revise our discussion about the spatial variation of the in situ AHe dates in Apatite-BaF for clarification, also taking into account suggestions from RC 2.

- **RC 2:** **When the data contradict the model, it typically indicates either low-quality measurements (which I assume is not the case here) or that the model itself may not be correct. Thus, this issue should be discussed in detail. Why did the classical He-loss model fail to reproduce the results? The authors provide only a brief discussion of this matter. Among the factors considered are the presence of inclusions, uranium-enriched zones, and variations in the degree of crystallinity. However, as it was already mentioned, profiles are reproducible, no inclusions are observed in the analyzed half of the grain, and the effect of apatite crystallinity on helium diffusion is limited. The possible influence of implanted helium is not addressed.**

- **AR 2:** Thank you for highlighting the issue of helium implantation. As for RC 1, we will address each aspect separately below.
  - o "Why did the classical He-loss model fail to reproduce the results?"
    We are not entirely sure what this question alludes to. Assuming the comment targets the counterintuitive in situ AHe date pattern, for which we admittedly do not have an explanation, we suggest further expanding the discussion and elaborating on the variability in crystal lattice properties and diffusivity not directly related to the parent nuclides but to variations in major element composition (e.g. Djimbi et al. 2015) or vacancy damage (e.g. Gerin et al. 2017), which we have neglected to mention.

However, with the information we have for Apatite-BaF, we cannot determine whether these factors are significant.

- o "The possible influence of implanted helium is not addressed."

  We thank the reviewer for pointing this out and recognise that a discussion of this topic should be added to the manuscript. We do not think that $^4$He implantation has a significant influence on the $^4$He profiles in Apatite-BaF for the following reasons:

  1. When evaluating the potential impact of implanted $^4$He from external sources, it is commonly assumed that the outer 20 μm rim of the apatite grain is affected the most (e.g. Spiegel et al., 2009; Gautheron et al., 2012). For our measurements, the distances of the in situ $^4$He ablation spots to the grain rims are larger than 20 μm, except for two in situ $^4$He ablation spots that were neither included in thermal modelling nor (U-Th-Sm)/He age calculations (refer to Fig. 2 and Table 3 in the manuscript for spot distances to the grain rim).

  2. Furthermore, models of implantation scenarios predict a significant peak in $^4$He concentration at the grain rim facing the external $^4$He source (see, for example, Fig. 6 in Gautheron et al. 2012). However, our Apatite-BaF $^4$He profiles do not show this peak, suggesting that the impact of $^4$He implantation is not significant.

  We will add a paraphrased version of the above elaboration to the discussion section of our manuscript for clarification.

- **RC 3: This point is of critical importance—if the model does not accurately describe observed in situ (U-Th-Sm)/He profiles, its validity as a basis for thermal history modeling becomes questionable.**

- **AR 3:** In our opinion, the modelling results for both Apatite-URG and Apatite-BaF clearly demonstrate that in situ $^4$He profiles can be inverted for plausible thermal histories (Fig. 5 and Fig. 6 in our manuscript). Given that zoned or heterogeneous grains such as Apatite-BaF are known to be problematic for conventional whole-grain (U-Th-Sm)/He approaches where rigorous criteria for optimal grain selection apply, we do not see the rationale in dismissing the in-situ $^4$He profile approach based on difficulties with interpreting the heterogeneous Apatite-BaF.

  In our opinion, the validity of the in situ approach hinges on the ability to produce accurate results for homogeneous grains (as for whole-grain (U-Th-Sm)/He dating). We want to emphasise here that the homogeneous Apatite-URG yielded good results, and we thus consider the method valid. Apatite-BaF, on the other hand, highlights the limitations and areas of further development.

  We suggest modifying our manuscript in several places to 1. explain our choice in analysing homogeneous and heterogeneous grains (testing the validity and limitations of the approach), 2. explicitly clarify that homogeneous grains are the most suitable and straightforward choice for routine analyses and 3. state that zoned or heterogeneous grains, such as Apatite-BaF, can be analysed, but are not recommended.

**General comments**

- **RC 4: For a study of this kind, it would have been preferable to use apatite grains from thermochronologically well-characterized localities, where FT and AHe ages are already available, or where even 4He/3He profiles have been**

**obtained. While this may not be essential from a methodological point of view, it would undoubtedly strengthen the testing and validation of the proposed thermochronological modeling approach.**

- **AR 4:** This is a valid criticism, and we agree with the reviewer that a well-characterised sample or standard material would have allowed for better comparability of our approach to whole-grain (U-Th-Sm)/He dating and 4He/3He analyses.

  Unfortunately, the well-characterised standard material available in our laboratory (Fish Canyon Tuff apatite (FCT) and Durango apatite) was not suitable for our measurement approach. For Fish Canyon Tuff, the available grains were too small. Further, zonation has been identified as a concern for this standard (e.g. Pickering et al. 2020). For Durango apatite, the grains were too large for the analysis to be performed within a reasonable cost and timeframe.

  For this reason, we chose Apatite-URG (from a Miocene tuff with an independently constrained U-Pb age) as a substitute for standard material. The Apatite-URG sample provided large, euhedral and clear crystals. Furthermore, the expected cooling history for Apatite-URG is simple and sufficiently well-constrained due to the volcanic nature of the source rock and the independently determined U-Pb age. Therefore, we consider this sample to be appropriate for testing and validating our methodology.

  We will add a few sentences explaining the choice of sample material to our manuscript.

- **RC 5: Sample preparation involved SEM imaging and mounting in Teflon, both of which exposed the samples to elevated temperatures. The authors likely used low current values for SEM imaging and low temperatures for mounting to limit possible thermal effects; thus, it should not affect the AHe ages. But detailed information regarding this issue should be presented in the methodological section.**

- **AR 5:** Thank you for pointing out the lack of discussion concerning the thermal effects of sample preparation. We will add the Teflon embedding temperature and duration (300°C for 2 minutes) as well as the SEM imaging settings (beam current 15 kV) in the methods section.
  - Concerning the potential effect of Teflon embedding, we can calculate the fractional He-loss resulting from such a short-term heating event using the approximation from Reiners et al. (2007) (Equation 3). With the diffusion parameters determined for the Durango apatite standard (Farley, 2000) as an estimate and using our measured grain radii, we can approximate that the fractional He-loss from Teflon embedding is ~1% for grains the size of Apatite-BaF and ~0.5% for grains the size of Apatite-URG. We will add this information to the methods section.
  - The impact of SEM analyses on the AHe system has been systematically examined by Shan et al. (2013). They found that AHe dates are not significantly affected by SEM. They also determined that the majority of heating caused by SEM would be limited to the upper 0.3 μm of the grain. We will refer to Shan et al. (2013) in the methods section.

- **RC 6: What are the reasons for the significant variations in He extraction pit depth observed within a single grain (for example, from 6.9 to 9.3 μm; approximately 25%; Table 3)? If the laser energy is dissipated/reflected, could it**

**potentially heat the sample (mobilize He) or heat internal parts of the cell (increase blank level)?**

- **AR 6:** We report the maximum pit depth in Table 3. Due to the unevenness of the pit floor, the maximum pit depth can vary significantly. However, the variation in maximum pit depth does not necessarily mean the ablated volume strongly varies (Figure 1). For example, in Apatite-URG, the maximum pit depth ranges from 6.9 to 9.3 µm, but the standard deviation of the mean pit volumes is only 4%. We suggest removing the maximum pit depth information from the manuscript and only reporting the mean pit depth.

  We are unsure what the second comment regarding the dissipation of laser energy, the mobilisation of He from heating of the sample and the increase in blank level allude to in the context of pit dimension variations. As for the heating of the grain outside the pit during He extraction, Boyce et al. (2006) reported that laser heating and He degassing outside the ablation pit are negligible.

[Figure]

*Figure 1: Cross-section through two exemplary ablation pits with significant variation in max. pit depth, but quite consistent pit volumes.*

**Additional minor comments can be found within the attached file.**

Comments from the attached pdf file

Note: The minor comments have been marked and annotated in the text and figures. We have copied the reviewer's comments and added the line information. For reference, we have also quoted text fragments marked by the reviewer and indicated which section of the manuscript the reviewer's comment concerns.

Abstract

- Line 21 ("for our study and instrument set-up"): **provide extra details (Resochron, eximer etc)**

- We will add "RESOchron system (Applied Spectra), consisting of a He-line and an excimer laser".

- Line 24 ("20-30microns"): **you tested only 20 microns, aren't you?**
  - In this sentence, we meant to make a recommendation for an optimal spot diameter. We will rephrase this sentence and add another sentence detailing the range of spot sizes tested (10 – 30 μm) and the respective grains' AHe dates.

- Line 24 ("six-spot in situ 4He profile requires a minimum grain diameter (measured perpendicular to the c-axis) of 145 μm"): **is it true for all apatites or only for the young ones?**
  - This is our general recommendation. With an optimal ablation spot diameter of 20 μm, a grain size of at least 145 μm is needed to measure a $^4$He profile composed of six individual in situ $^4$He measurements. (Details are in section 4.1 of our manuscript.) We will modify this sentence to clarify this, e.g. by writing " Additionally, **with a $^4$He ablation spot diameter of 20 microns**, a six-spot in situ $^4$He profile requires a minimum grain diameter (measured perpendicular to the c-axis) of 145 μm".

- Line 26 ("potentially including zoned and irregularly shaped crystals"): **so you suggest that BaF -like apatite can be used for modeling?**
  - Yes, we suggest that heterogeneous "BaF-like" apatites can be used for modelling.

Methods

- Line 85 ("age"): **U-Pb**
  - Thank you for pointing this out. We will add U-Pb before "age standard" for clarification in line 85.

- Line 90 ("embedded in a Teflon sheet"): **At what temperature?**
  - The hotplate used for embedding the grains in Teflon was set to 300°C, and the mount remained on the hotplate for 2 minutes. We will add this information.

- Line 92 ("SEM"): **SEM heat the grains. What energies have you used? What was the beam current? Were the samples coated? By gold or by coal?**
  - We did not coat the sample mount. For the SEM scan, we used a voltage of 15 kV. The emission current was 40.3 μA, the filament power was 4.46 W, and the dwell time was 200ns. We will add this information to the manuscript.

- Table 1 **Which coordinate system? WGS84 or ETRS89?**
  - We use WGS84. We will add this information to the table notes.

- Line 108 ("Line blanks were recorded regularly"): **It is cold blank. What about the hot blank? Have you measured the blank when ablate helium free sample?**

- Each sample mount is individually placed in an ultra-high-vacuum (UHV) laser cell and evacuated for a minimum of 24 hours to ensure that vacuum levels reach the low-pressure regime required for high-precision He analysis. In fact, our cold cathode was unable to measure the pressure since it reached values below the detection limit 1E-8 torr. Before analysing samples, several initial blank analyses are run until blank $^4He/^3He$ ratios reached values <0.0001. These blanks use the identical valve and volume configuration as the subsequent sample measurement, but without firing the laser.

- Line 109 ("0.0003 to 0.0005 ncc"): **How do you calibrate the MS?**
  - Quantification of radiogenic $^4He$ extracted from individual laser pits relies on calibration against an internally monitored 4He reference reservoir ("Q-tank"). The $^4He$ volume of the Q-tank is known but not constant: it undergoes predictable exponential depletion as Q-shots are consumed during routine analytical work. To maintain accuracy, the Q-tank is recalibrated at regular intervals, typically annually or after several thousand Q-shots, using a secondary reference reservoir ("D-tank"). The D-tank contains an independently certified amount of $^4He$, externally calibrated by Des Patterson, and serves as the long-term stability anchor for all Q-tank recalibrations. Calibration factors derived from these intercomparisons are used to convert measured ion currents into absolute $^4He$ amounts in sample analyses.

- Line 111 ("after blank correction"): **have you incorporated the blank uncertainty into standard deviation calculations?**
  - Yes, blank uncertainty is propagated into the $^4He$ concentration standard deviation. We will report the equations in the supplementary material.

- Line 122 ("re-polished grain surfaces"): **You have removed 10 microns or more?**
  - Thank you for this question. From the pit depths, we can estimate that ~10 µm had to be polished away to smooth the grain surface. We were careful in repolishing the mount and checked the state of removal multiple times during the process to make sure we stopped polishing as soon as the pit traces had been removed. We will expand the methods section and provide more details about the repolishing procedure, e.g. by adding **"Prior to parent nuclide measurements, the grains were polished for 3.5 h on a polishing machine at intervals of 4 to 20 min with a decreasing force from 20 to10 N to remove the helium extraction pits and create an even surface for U, Th and Sm distribution mapping. The state of removal was checked multiple times during the process, and polishing was stopped as soon as all visible pit traces had been removed."**

- Line 124 ("Durango"): **Durango?**
  - We will add "age standard for apatite" and the reported standard (U-Th)/He age in parentheses for clarification.

- Table 2 ("24"): **You have a large variations in pit depths for He measurements, and quite a constant values for trace elements. Why?**

- o Thank you for this question. We observe variations in the depths of the $^4$He ablation pits, but not in the depths of trace element pits, as the former are individually measured, while the latter is a calculated value. We measured the pit dimensions of each $^4$He ablation pit individually because the pit volume is necessary for $^4$He concentration calculation (lines 111 – 114 in the manuscript). The pit depth for the trace element measurements, on the other hand, is a calculated value that we derived from the established ablation-time-depth relationship (lines 129 – 130 in the manuscript). For the trace element measurements, our interest only lies in the depth in the grain from which the signal originates, not in the exact pit dimension. Hence, the calculated value suffices. We will clarify this in the table notes. We will also replace "Mean spot depth" with "Mean pit depth" in the header of Table 2 (and where relevant throughout the manuscript) and add the $^4$He pit depth SD for clarity.

3. Results

- Line 213 ("of 2.8E15 ± 5.0E15 at g-1 to 8.5E15 ± 2.5E15 at g-1"): **why haven't you used 20 microns spot for McClure? It would allow you to receive a profile, wouldn't it?**
    - o The reviewer raises a good point. We wanted to test different $^4$He ablation spot sizes to test the limitations of the in situ $^4$He measurements concerning spot size, and thus used a different ablation spot diameter for each analysed grain. We anticipated a higher $^4$He concentration in Apatite-McClure and therefore chose to test our smallest spot diameter in this grain. We fully agree with the reviewer that, in hindsight, a bigger spot would have been the better choice.

- Line 250 ("Apatite-McClure displays minor internal variation (…)"): **McClure apatite is well-described. Given that this data is not used in modelling I would suggest it be redundant**
    - o While Apatite-McClure highlights the limitations of our method, especially concerning ablation spot size, we agree that it is of minor importance compared to Apatite-URG and Apatite-BaF. We suggest removing detailed descriptions concerning Apatite-McClure and moving the remaining information that is relevant to the manuscript to the appendices.

- Line 253 ("slight (…) 5-17 ppm"): **slight? 3 times difference**
    - o Both the U-concentration and the variation (5-17 ppm) are small compared to Th (86–234 ppm) and Sm (20–310 ppm). We will rephrase this sentence to clarify that the total amount and variation in U is small or "slight" relative to Th and Sm.

- Line 283 ("83.22 ± 9.55 Ma to 162.25 ± 28.95 Ma"): **round the values appropriately**
    - o Thank you for pointing this out, we will do that throughout the manuscript.

- Line 293 ("was 20 Ma"): **why not 16.75, or 16.75+0.84?**
    - o We concede that the model box is larger than strictly necessary. We wanted to avoid the starting point dictating the inversion result. Given that the U-Pb and (U-Th-Sm)/He dates for Apatite-URG overlap, we drew a larger exploration box, including

up to 20 Ma. Ultimately, this was larger than necessary, but as it does not influence the AHe date, it is a question of presentation.

- Line 305("mean apatite U-Pb date derived from trace element measurements in Apatite-BaF"): **add the U-Pb data as a supplementary file**
  - The U-Pb data are already provided in the data supplement linked on Zenodo. We will refer to the supplement with a link in the text.

- Line 306 ("near the sample location"): **where exactly? 1 km - 10 km distance? same geological unit?**
  - Thank you for this comment. Unfortunately, we can only estimate the distance of our Apatite-BaF sample location to the sampling sites of Vamvaka et al. (2014) since they do not provide coordinates, but only an overview map.
    We estimate that the distance between our Apatite-BaF sample location and the nearest sampling site of Vamvaka et al. (2014) is approximately 5-10 km. For this nearest sample, they report that the sampled rock type is granite, which aligns with the lithology of our sample. However, we cannot be certain that we have sampled the same intrusion.
    Vamvaka et al. (2014) suggest that an exhumation-and-reheating scenario is generally plausible for the Bavarian Forest. Our intention with the comparison of our Apatite-BaF to findings from Vamvaka et al. (2014) was to assess the plausibility of the derived cooling history for Apatite-BaF, not to prove its accuracy. We will emphasise this in Section 3.5.1.

- Fig.5 and Fig. 6: **what are those artefacts near the center?**
  - Thank you for this important question. We will expand Section 2.5 to clarify the reasons for the artefacts, and we will also add a brief explanation to the figure captions of Figs 5, 6, and 7.
    The forward-modelled $^4$He profiles display a "jump" near the centre, because they merge two core-rim profiles. In theory, for a perfectly homogeneous grain, only one core-rim profile would be needed, since both halves of the grain (from core to rim) would yield identical information. The forward model we used (detailed in Glotzbach & Ehlers, 2024) is designed for displaying such core-rim profiles.
    To model a full rim-rim profile, which is vital for asymmetric $^4$He profiles (heterogeneous grains), we combine two core-rim profiles. Since both core-rim profiles share a common point at the grain centre, this centre point would be defined twice in the model. Therefore, it must be excluded (left undefined) in one of the profiles. As a result, the concentration appears to "jump" where the profiles meet.

- Line 332 as well as one apatite grain from the McClure Mountain Syenite standard (Apatite-McClure): **no 4He profiles for McClure**
  - We will remove information about Apatite-McClure from this section.

- Line 335 ("The measurement uncertainties for Apatite-McClure (ablation spot diameter 10 μm) exceed 40%"): **it is not a main result, i guess**
  - We will remove the information about Apatite-McClure from this section.

- Line 346 ("sensitivity testing"): **new paragraph**
  - We will add a new paragraph.

- Line 349 ("the high individual measurement uncertainties for 4He"): **The used ablation pit was too small. Nothing wrong with McClure - 4He concentrations are high enough. Please, rephrase**
  - We will remove the sentences referring the Apatite-McClure from this section.

- Line 355 ("and heterogeneous grains"): **part of heterogeneous?**
  - We are not entirely sure what the comment intends to point out here. We suggest rephrasing the sentence for better clarity, e.g. "In situ $^4$He profiles can be inverted for cooling histories of homogeneous and, even though more challenging, heterogeneous grains."

Discussion

- Line 376 ("SD<5%"): **see table 3, SD is > 5%, while 3 blanks level is achieved**
  - We thank the reviewer for this comment. The sentence is poorly worded, and we will rephrase it.
    We ideally want a signal clearly above 3x the blank level with a standard deviation (SD) <5%. For our instrument set-up, measurements that are only slightly above 3x the blank level typically have blank-corrected SDs around 6-7%. Generally, the lower the difference between the signal and the blank, the higher the uncertainty in the measurements.

- Line 423 ("older dates towards the grain rim and younger dates towards the grain centre"): **It is a bit misleading, as URG might not have this trend (your next sentence)**
  - We will rephrase this section and shorten it by removing most references to Apatite-URG. We will only discuss Apatite-BaF in detail.

- Line 429 ("In both cases, the observed date distribution within the grains is counterintuitive"): **You don't have data to proof this trend for URG sample.**
  - We will remove sentences discussing Apatite-URG.

- Line 437 ("From our data, we cannot decipher the reason for the observed inverted in situ date pattern"): **this should be discussed in the detail! what is the most likely scenario? what about alpha implantation? whats is an average U and Th content of a host rock?**
  - We will add elaboration on the impact of $^4$He implantation as per our detailed response to RC 2.

- Fig. 7 **why near the zero value do you have this mathematical artefact**

- o   Please refer to the above explanation for Figs. 5 and 6.

- Line 510 ("accurately"): **the error is quite large**
  - o   We agree with the reviewer's point that the measurement uncertainty is relatively large. Although it may be a semantic question, it is crucial to distinguish that larger measurement uncertainties do not imply inaccuracy, but rather imprecision. The thermal histories we derive from the $^4$He profiles for Apatite-URG all show very rapid cooling in the Miocene, which aligns with expectations for a volcanic rock, such as the source rock of Apatite-URG. Furthermore, the timing of cooling is consistent with the independently determined U-Pb age. Thus, we maintain that the derived cooling history is accurate, even though the $^4$He profile measurements are arguably imprecise.

**Conclusion**

- Line 558: delete "Using"
  - o   Ok, we can delete the word.

**References**

Boyce, J. W., Hodges, K. V., Olszewski, W. J., Jercinovic, M. J., Carpenter, B. D., & Reiners, P. W. (2006). Laser microprobe (U-Th)/He geochronology. *Geochimica et Cosmochimica Acta*, *70*(12), 3031–3039. https://doi.org/10.1016/j.gca.2006.03.019

Djimbi, D. M., Gautheron, C., Roques, J., Tassan-Got, L., Gerin, C., & Simoni, E. (2015). Impact of apatite chemical composition on (U-Th)/He thermochronometry: An atomistic point of view. *Geochimica et Cosmochimica Acta*, *167*, 162–176. https://doi.org/10.1016/j.gca.2015.06.017

Farley, K. A. (2000). Helium diffusion from apatite: General behavior as illustrated by Durango fluorapatite. *Journal of Geophysical Research: Solid Earth*, *105*(B2), 2903–2914. https://doi.org/10.1029/1999jb900348

Gautheron, C., Tassan-Got, L., Ketcham, R. A., & Dobson, K. J. (2012). Accounting for long alpha-particle stopping distances in (U-Th-Sm)/He geochronology: 3D modeling of diffusion, zoning, implantation, and abrasion. *Geochimica et Cosmochimica Acta*, *96*, 44–56. https://doi.org/10.1016/j.gca.2012.08.016

Gerin, C., Gautheron, C., Oliviero, E., Bachelet, C., Mbongo Djimbi, D., Seydoux-Guillaume, A. M., Tassan-Got, L., Sarda, P., Roques, J., & Garrido, F. (2017). Influence of vacancy damage on He diffusion in apatite, investigated at atomic to mineralogical scales. *Geochimica et Cosmochimica Acta*, *197*, 87–103. https://doi.org/10.1016/j.gca.2016.10.018

Glotzbach, C. and Ehlers, T. A. (2024). Interpreting cooling dates and histories from laser ablation in situ (U–Th–Sm)∕He thermochronometry: a modelling perspective, Geochronology, 6, 697–717, https://doi.org/10.5194/gchron-6-697-2024

Pickering, J., Matthews, W., Enkelmann, E., Guest, B., Sykes, C., & Koblinger, B. M. (2020). Laser ablation (U-Th-Sm)/He dating of detrital apatite. *Chemical Geology*, *548*(March), 119683. https://doi.org/10.1016/j.chemgeo.2020.119683

Reiners, P. W., Thomson, S. N., Mcphillips, D., Donelick, R. A., & Roering, J. J. (2007). Wildfire thermochronology and the fate and transport of apatite in hillslope and fluvial environments. 112. https://doi.org/10.1029/2007JF000759

Shan, J., Min, K., & Nouiri, A. (2013). Thermal effects of scanning electron microscopy on He diffusion in apatite : Implications for ( U \ Th )/ He dating. *Chemical Geology*, *345*, 113–118. https://doi.org/10.1016/j.chemgeo.2013.03.001

Spiegel, C., Kohn, B., Belton, D., Berner, Z., & Gleadow, A. (2009). Apatite (U-Th-Sm)/He thermochronology of rapidly cooled samples: The effect of He implantation. *Earth and Planetary Science Letters*, *285*(1–2), 105–114. https://doi.org/10.1016/j.epsl.2009.05.045

Vamvaka, A., Siebel, W., Chen, F., & Rohrmüller, J. (2014). Apatite fission-track dating and low-temperature history of the Bavarian Forest (southern Bohemian Massif). *International Journal of Earth Sciences*, *103*(1), 103–119. https://doi.org/10.1007/s00531-013-0945-x

---

## Author Comment (AC2)

**Authors' response to Review 2**

- Reviewer: Julien Amalberti
- Citation: https://doi.org/10.5194/egusphere-2025-3879-RC2

Note: The reviewer comments are presented in black print in their original, unabridged wording. The author's response is displayed in blue.

**Review of Maier et al. (submitted to Geochronology)**

This manuscript describes a recent effort to develop in situ (U-Th-Sm)/He analytical methods and a modeling strategy applied to single apatite grains. The paper builds upon previous work by Danišík et al. (2017) and Glotzbach & Ehlers (2024), exploring specific analytical parameters such as ablation laser spot density, size, and location. The authors use natural apatite samples with both homogeneous and heterogeneous U-Th-Sm concentrations to evaluate their proposed in situ (U-Th-Sm)/He strategy. The manuscript also details the methodology used for both 4He measurement and parent nuclide analysis.

Results indicate that 4-6 laser spots, ranging from 20-30 µm in diameter, are sufficient to derive a 4He profile from 16 Ma apatite grains of approximately 145 µm. The authors recommend these parameters as the minimum analytical requirement for in situ (U-Th-Sm)/He analyses. They also combine 4He profiles along the c-axis (3-4 profiles per grain) with 3D reconstructions of radiogenic parent nuclide distributions to improve the accuracy of modelled thermal histories.

Overall, the paper is well written and easy to follow, but some sections lack sufficient methodological detail (see comments below). The methodology section would benefit from clearer descriptions to remove potential biases in the extracted He dataset. Moreover, the modeling approach would have been strengthened by including a case study using a well-characterized apatite standard (e.g., Fish Canyon Tuff or Durango) to test the validity of the authors' strategy.

In addition, the study decouples the He pit locations from the parent isotope analyses by re-polishing the surface (of unknown thickness) between the 4He and U-Th-Sm analyses. To my knowledge, previous studies tend to perform parent isotope ablations within the same pits as 4He measurements (e.g., Pickering et al., 2020) to avoid losing spatial information during re-polishing. It would have been useful if the authors provided more justification for this approach (see comments below).

Nevertheless, this study represents a meaningful advancement in in situ (U-Th-Sm)/He methodology, particularly with the introduction of the new Caw parameter for parent nuclide correction at 4He ablation sites, which accounts for surrounding α-ejection effects. In my opinion, this is the most novel and valuable aspect of the study, as it tries to addresses 4He implementation influenced by α-ejection from zones of differing eU within the grain (see

Pickering et al., 2020, Fig. 8). Therefore, I consider this manuscript a valuable contribution to the advancement of in situ (U-Th-Sm)/He thermochronology.

Below I provide my main comments, followed by detailed comments. In summary, the paper is well constructed and represents a significant contribution, but requires clarification of several methodological aspects (particularly data acquisition and error handling) and would benefit from comparison to a well-known system for validation. I therefore recommend this manuscript for publication after revision.

We thank the reviewer for their time and careful review of our manuscript. The reviewer's main concern is a lack of sufficient methodological detail. Furthermore, they express the need for a better justification of methodological choices, particularly with respect to the re-polishing step between $^4$He and parent nuclide measurements. Lastly, they point out the lack of analysis of a well-characterised standard material. We address these concerns in our answers to the main and detailed comments below. We are optimistic that our answers and suggested revisions to the manuscript will alleviate the reviewer's concerns.

We have numbered the reviewer comments (RC) and author responses (AR).

**Main Comments**

- **RC 1: Lack of comparison with a well-known system:** Although not critical, the study would be considerably strengthened by comparison with a well-characterized apatite system such as Fish Canyon Tuff (FCT) or Durango. FCT apatite, in particular, would be an excellent candidate given recent 4He/3He data from the distal (FCT-D) and classic (FCT-C) localities (Colleps et al., 2024). These samples provide a well-constrained framework (Renne et al., 1998; Reiners et al., 2002; Phillips & Matchan, 2013; Gleadow et al., 2015) encompassing both non-diffusive alpha-ejection-only (FCT-D) and more diffusive post-eruptive burial reheating (FCT-C) apatite behavior. Using such a reference material could effectively validate the authors' analytical and modeling strategy.

- **AR 1:** We agree with the reviewer that analysing a well-characterised standard material such as Fish Canyon Tuff (FCT) or Durango would have greatly improved the direct comparability of our approach to whole-grain (U-Th-Sm)/He and $^4$He/$^3$He analyses.
  We did not analyse FCT or Durango, because the FCT grains available at our laboratory were too small for our analytical approach, while our Durango apatite material was too large to be analysed within a reasonable time and cost investment. Furthermore, FCT has been shown to exhibit significant parent nuclide zonation (e.g. Pickering et al. (2020) which disqualified it from being good validation material for our purposes.
  Against this background, we chose to use the Apatite-URG sample as validation material for our in situ approach, because it contained reasonably large, euhedral, clear and homogeneous crystals. Additionally, Apatite-URG (being sourced from

Miocene foiditic tuff and with an independently determined apatite U-Pb age (Binder et al., 2023) has a sufficiently simple and well-constrained expected cooling history. We suggest modifying Section 2.1 "Samples and Sample preparation" to explicitly explain and justify our choice in sample material and address the absence of well-characterised standards with the reasoning provided above.

- **RC 2: Decoupling of 4He and parent nuclide locations:** The authors performed parent nuclides measurements after re-polishing the apatite surface, but the thickness removed is not specified. I am concerned that their 3D reconstructions, built from an unknown removed material layer below the 4He spots, may miss critical information. A simple addition of analyzing parent nuclides directly within or immediately below the 4He pits could have provided a valuable internal check. These data would allow the authors to test whether their calculated Caw values are consistent with measured parent concentrations. Clarification and justification of this methodological choice are needed.
- **AR 2:** We thank the reviewer for this comment. We will expand Section 2.3 "Parent nuclide mapping" to include necessary details and justification for the repolishing step. Additionally, we will add annotations and a better depiction of this analytical step in Fig. 1.
  - We repolished the grain surface after the $^4$He measurement to map the distribution of parent nuclides across the entire exposed internal grain surface, rather than only obtaining concentrations at the $^4$He ablation sites. While it is theoretically possible to avoid repolishing the grain surface and instead measure the concentration of the parent nuclide in and around the $^4$He ablation pits, doing so in practice presents challenges. Parent nuclide signals would originate from different depths within the grain (because the measurements do not start in a flat plane). This would require us to correct for ablation depth when constructing parent nuclide maps, because parent nuclide measurements from around the $^4$He ablation pits would not reach the same depth in the grain as measurements conducted inside the $^4$He pits, introducing another source of uncertainty.
  - To polish away the visible $^4$He pit traces, we needed to remove a maximum of 10 μm, based on the maximum depth of the deepest pits. We were very careful during re-polishing and checked the state of the removal regularly to avoid removing too much material. Our re-polishing protocol was as follows: "Prior to parent nuclide measurements, the grains were polished for 3.5 h on a polishing machine at intervals of 4 to 20 min with a decreasing force from 20 to 10 N to remove the helium extraction pits and create an even surface for U, Th and Sm distribution mapping. The state of removal was checked multiple times during the process, and polishing was stopped as soon as all visible pit traces had been removed." We will add this information to the methods section.

- RC 3: **Methodological transparency:** Several detailed comments below relate to blanks, uncertainties, and analytical procedures. Although the authors provide substantial data, key aspects of measurement uncertainty remain unclear, particularly where modeling results appear counterintuitive. Providing additional information on blank correction and error propagation would significantly improve the reproducibility and credibility of the dataset.
- AR 3: We will add the requested information.

**Detailed Comments**

- RC 4: **Line 101:** The authors mention acquisition of a 4He profile parallel to the c-axis, yet no such data are presented. Did this profile yield similar results to the perpendicular one? If so, it should be shown or discussed.
- AR 4: Thank you for bringing this to our attention. The data for the c-axis parallel profile are available in the supplementary material. We will refer to this in the text. We will also add sentences to the methods section explaining the reason for measuring the c-axis parallel profiles.
  - We measured a c-axis parallel profile to verify whether the $^4$He concentrations were consistent in the grain along this direction. In retrospect, we realised the same information can easily be obtained by comparing the perpendicular profiles, making the c-axis parallel measurements superfluous. Hence, we don't display this data in the manuscript.
  - Geometric considerations for $^4$He diffusion are different for c-axis parallel profiles. Thus, we did not attempt thermal modelling.

- RC 5: **Line 109:** 4He blanks are given in ncc, whereas Table 3 and Appendix B use atoms/g for 4He measurements. I strongly recommend using consistent units to avoid conversion errors. Reporting blanks in atoms or normalizing to ccSTP would be most appropriate.
- AR 5: We can convert the blanks to atoms. As atoms/g is the unit used for our inverse and forward modelling, we will continue using this unit for the tables and figures in the manuscript. The $^4$He measurements are already reported both in ncc and in atoms/g in the supplementary data. We will refer to the supplementary material in the text.

- RC 6: **Line 111:** The blank correction error for successful 4He measurements is specified as 6–15%, corresponding to apatite URG and BaF in Table 2. The blanks reported by the authors range from 0.0003 to 0.0005 cc (or 8.07×106 to 1.35×107 atoms), yielding an average blank of 1.08×107 atoms ± 2.7×106 atoms (or 0.0004 ± 0.0001 cc). Assuming no additional error from the 4He content itself (not provided in the dataset), the propagated error based solely on the blank value would be approximately 14% for Ap-URG and 7–10% for Ap-BaF. Including

the pit volume errors reported in Table 2 (~5% for URG and ~9% for BaF) increases the total 4He concentration uncertainty to roughly 15% for URG and 11–14% for BaF. These estimates, based on the average blank and its uncertainty using a weighted blank contribution factor (i.e., signal/blank ratio), are close to the authors' stated range of 6–15%, and even exceed the maximum 4He error reported for apatite BaF (<6%). This is without considering any analytical uncertainty associated with the laser ablation measurements. I therefore question how the authors derived their total error estimates. Furthermore, published RESOchron systems typically report 4He blank values between $3.2 \times 10^7$ atoms (0.0012 ncc; Evans et al., 2015) and $6.7 \times 10^7$ atoms (0.0025 ncc; Danišík et al., 2017), whereas the authors report significantly lower blanks ($8 \times 10^6$ to $1 \times 10^7$ atoms, or 0.0003–0.0005 ncc). While I acknowledge that each analytical setup is unique and background levels can vary, it would be helpful if the authors provided additional details about their analytical protocol that could explain these notably lower 4He blank values. Finally, the authors report 1σ uncertainties on the measured in situ AHe ages ranging from 10% to 21% for both URG and BaF, yet my own propagation of the reported errors on 4He, 238U, 232Th, and 147Sm value (Table 3) yields roughly twice those values. It would therefore be useful if the authors detailed their analytical setup and blank correction procedure to explain these unusually low blank values and how final age uncertainties (10–21%) were propagated.

- **AR 6:** We have rechecked our blank levels and reviewed our calculations. We continue to obtain the same measurement uncertainties as reported in our manuscript. Below, we provide details on our analytical setup and the error propagation. We will include this information as a supplementary file to our manuscript.

- **In-situ (U-Th)/He Analytical Procedure**
  1. Mass Spectrometer Calibration
     Quantification of radiogenic $^4$He extracted from individual laser pits relies on calibration against an internally monitored $^4$He reference reservoir ("Q-tank"). The $^4$He volume of the Q-tank is known but not constant: it undergoes predictable exponential depletion as Q-shots are consumed during routine analytical work. To maintain accuracy, the Q-tank is recalibrated at regular intervals, typically annually or after several thousand Q-shots, using a secondary reference reservoir ("D-tank"). The D-tank contains an independently certified amount of $^4$He, externally calibrated by Des Patterson, and serves as the long-term stability anchor for all Q-tank recalibrations. Calibration factors derived from these intercomparisons are used to convert measured ion currents into absolute $^4$He amounts in sample analyses.

  2. Sample Loading and Preparation

Each sample mount is individually placed in an ultra-high-vacuum (UHV) laser cell and evacuated for a minimum of 24 hours to ensure that vacuum levels reach the low-pressure regime required for high-precision He analysis. In fact, our cold cathode was unable to measure the pressure since it reached values below the detection limit 1E-8 torr. Before analysing samples, several initial blank analyses are run until blank $^4$He/$^3$He ratios reached values <0.0001. These blanks use the identical valve and volume configuration as the subsequent sample measurement, but without firing the laser.

3. Measurement Sequence Design
Because Q-shots yield extremely stable $^4$He/$^3$He ratios (<1% variability) and analytical sequences are comparatively short (typically 20–40 analyses), each sequence begins and ends with two Q-shots. No Q-shots are included between samples to avoid flushing the line with high amounts of $^4$He and potential memory effects. The $^4$He content of Q-shots is 3–4 orders of magnitude higher than the He liberated during in-situ laser extraction. For this reason, the initial Q-shots are followed by 3–4 blank measurements to allow the system to return to pre-Q-shot blank levels. Sample measurements only begin once these blank values stabilize back to the initial baseline.

Blanks are measured frequently throughout the sequence. For the first three samples, a blank is run immediately before each measurement. As the system stabilizes, the number of sample analyses between blanks is gradually increased (from one, to two, three, four, and finally up to five), while maintaining sufficient blank coverage for accurate correction.

4. Blank Characterization and Correction
For each blank and sample measurement, $^4$He/$^3$He ratios are calculated. During the Apatite-URG and Apatite-BaF analytical sessions, mean blank ratios and standard deviations were:
- 0.0000677±0.0000034 (Apatite-URG)
- 0.0000700±0.0000016 (Apatite-BaF)

With a Q-standard $^4$He content of 10.69 ncc for these sessions, these ratios correspond to mean blank $^4$He amounts of approximately:
- 0.000700±0.000046 ncc (Apatite-URG)
- 0.000757±0.000048 ncc (Apatite-BaF)

Blank correction for each sample is performed by subtracting the pre-sample blank $^4$He/$^3$He ratio from the sample's measured ratio. This ensures that sample-derived radiogenic $^4$He is isolated from procedural background contributions, and that time-dependent drift in blank characteristics does not introduce bias into the final (U-Th)/He age calculations.

5. Helium error propagation
All measured masses are measured during 10 individual cycles (i) that last in total roughly 50 sec, and the ratio (R43) between masses 4 (M4) and 3 (M3) of each run is calculated with:

$$R43_i = \frac{M4_i}{M3_i}$$  Eq. 1

The mean and absolute standard deviation (SD) of the R43 measurement is calculated with:

$$\overline{R43} = \frac{1}{N}\sum_{i=1}^{N} R43_i$$  Eq. 2

$$SD_{R43} = \sqrt{\frac{1}{N-1}\sum_{i=1}^{N}(R43_i - \overline{R43})^2}$$  Eq. 3

The $^4$He content of Q-shot j (number of Q-shot) is derived from the depletion factor (DF = 0.999899832 with SD = 4.26343E-6) and the $^4$He content (11.098 ncc with SD = 0.011 ncc) determined for Q-shot 1000 with the following equation:

$$^4He_{Q_j} = {}^4He_{Q_{442}}DF^{(j-442)}$$  Eq. 4

Accordingly, the absolute standard deviation is:

$$SD_{{}^4He_{Q_j}} = {}^4He_{Q_i}\sqrt{\left(\frac{SD_{{}^4He_{Q_{442}}}}{{}^4He_{Q_{442}}}\right)^2 + \left((j-1000)\frac{SD_{DF}}{DF}\right)^2}$$  Eq. 5

To calculate the $^4$He content of a sample, we have to know the $^3$He content added to the sample. This is estimated based on the $^4$He of nearby Q-shots ($^4He_{Qj}$). Accounting for a drift during measuring time we fit a spline-function to $^4He_{Qj}$. We used a Monte-Carlo approach to take into account the uncertainty of the $^4He_{Qj}$, and randomly sampled N-times $^4He_{Qj}$ data from a normal distribution (using the mean $^4He_{Qj}$ and corresponding standard deviations). Afterwards, we evaluate the N-times fitted functions at the measuring time of our samples and the resulting $^4He_{Qt}$ and SD at measuring time are:

$$^4He_{Qt} = \frac{1}{N}\sum_{i=1}^{N} f_i(t)$$  Eq. 6

$$SD_{{}^4He_{Qt}} = {}^4He_{Qt}\sqrt{\frac{1}{N-1}\sum_{i=1}^{N}(f_i(t) - {}^4He_{Qt})^2}$$  Eq. 7

Blank ratios $R_{bl}$ are calculated with Eq. 1 and used to calculate blank-correct R43 sample measurements ($R_{Sbc}$) with:

$$R_{Sbc} = \frac{1}{N}\sum_{i=1}^{N}(R_i - R_{bl_i})$$  Eq. 8

Accordingly, the absolute standard deviation is:

$$SD_{R_{Sbc}} = R_{Sbc}\sqrt{\left(\frac{SD_{RS}}{R_S}\right)^2 + \left(\frac{SD_{Rbl}}{R_{bl}}\right)^2}$$  Eq. 9

Where the sample and blank ratios and standard deviations have been calculated with Eq. 2 and 3. Finally, the $^4$He content of a sample at time t is calculated with:

$$^{4}He_{Sbc_t} = R_{Sbc} \frac{^{4}He_{Q_t}}{R_{Q_t}}$$

Eq. 10

Accordingly, the absolute standard deviation is:

$$SD\,_{^{4}He_{Sbc_t}} = \,^{4}He_{Sbc_t} \sqrt{\left(\frac{SD_{R_{Sbc}}}{R_{Sbc}}\right)^2 + \left(\frac{SD\,_{^{4}He_{Q_t}}}{^{4}He_{Q_t}}\right)^2 + \left(\frac{SD_{R_{Q_t}}}{R_{Q_t}}\right)^2}$$

Eq. 11

4He concentration and uncertainty:
We divide the $^{4}$He content ($^{4}He_{Sbc_t}$) by the ablation pit volume ($V_{Pit}$) to obtain the $^{4}$He concentration ($C_{He}$). Finally, the pit volume uncertainty ($SD_{VPit}$) is propagated into the total measurement uncertainty for the $^{4}$He concentration ($SD_{CHe}$) as follows:

$$SD_{CHe} = C_{He} \sqrt{\left(\frac{SD\,_{^{4}He_{Sbc_t}}}{^{4}He_{Sbc_t}}\right)^2 + \left(\frac{SD_{VPit}}{V_{Pit}}\right)^2}$$

Eq. 12

- **AHe age calculation and uncertainties:**
  To calculate the AHe age and uncertainty we use the equations for the non-iterative solution to the age equation by Meesters & Dunai (2005).

- **RC 7: Line 122:** The re-polished thickness should be specified. By how much was the surface re-polished before parent isotope analysis?
- **AC 7:** We will add the requested details to the text. Please refer to **RC 2/AR 2,** where we have already elaborated in the repolishing procedure for detailed information.

- **RC 8: Line 178:** The authors note they cannot measure ablation spots within 40 μm of the rim due to potential 4He loss. This limitation could strongly affect the ability to model cooling histories, since rim regions often contain critical diffusional information (as in 4He/3He methods). Have the authors evaluated the potential impact of this constraint on slow-cooled samples?
- **AR 8:** We agree with the reviewer that the inability to calculate $C_{aw}$ <40 μm from the grain rim is a major drawback of our approach.
  - We would like to note that the sentence in line 178, "*we chose not to calculate $C_{aw}$ for $^{4}$He ablation spots with centres <40 μm to the grain rim,*" refers only to $C_{aw}$ calculation and consequently the ability to use the measurements for inverse modelling. We did not mean to state that we cannot or did not measure $^{4}$He concentrations <40 μm from the grain rim. We recognise that this is worded confusingly in several places in the manuscript, and we will revise the relevant sections.

- o Although we cannot utilise the spots that are <40 μm from the grain rim in the inversion for thermal histories, we can still use them for misfit calculation and comparison between measured and forward-modelled $^4$He profiles to ensure the inversion results are reasonable. We thus still use information from the rim region. We will emphasise this in the discussion and suggest adding annotations to both Figs. 2 and 6 to distinguish which spots were used for the inversion and which were used for forward modelling/misfit calculation.
  - o For further discussion on thermal history modelling, please refer to RC 10/AR 10.

- **RC 9: Lines 209-214:** I am curious where the large 4He errors (>40%) originate. Based on the reported concentrations, apatite McClure appears to have 4He concentration ranging between the other two apatite grains, suggesting that each ablation pit should provide sufficient 4He for reliable analysis. However, the laser spot size for McClure (10 μm) is the smallest among the samples (BaF: 20 μm; URG: 30 μm), which could potentially explain the high uncertainties if the ablated volume, and thus the total 4He signal, was too low despite appropriate concentrations. Nevertheless, according to Appendix B (Table B1), the 4He concentrations measured in individual pits fall within the same range as those for URG and BaF. While the latter two samples display relatively low uncertainties (see also my previous comments regarding error propagation), apatite McClure exhibits unusually large errors. These discrepancies cannot be explained solely by 4He concentration or laser spot size, as the pit-by-pit 4He concentrations for McClure are intermediate between URG and BaF. In addition, the pit measurement error is reported at approximately 10% for 10μm laser spot size, which cannot account for the >40% uncertainties reported for McClure 4He concentrations. Could the authors clarify how these large errors were derived and identify the main contributing factors?
- **AR 9:** The large uncertainties for $^4$He measurements in Apatite-McClure stem from the too small ablated volume, which in turn resulted in a too low $^4$He signal. As the reviewer correctly noted, the $^4$He concentration in Apatite-McClure is comparable to Apatite-URG. However, the ablation pit volume for Apatite-McClure was only ~7% of the ablation pit volume for Apatite-URG (see Table 2 in our manuscript). Keeping in mind that the $^4$He concentration is the amount of $^4$He in a specific volume (of apatite that was ablated), it follows that if the ablated volume is small, the $^4$He content will also be small (assuming a constant $^4$He concentration).
Consequently, the $^4$He signal that we measured for Apatite-McClure was much lower than that for Apatite-URG, even though the concentrations are comparable. In fact, for Apatite-McClure, the $^4$He signal was so low that it was not significantly different from the blank level. Hence, the associated measurement uncertainty is very large. Kindly refer to AR 6 for details on error propagation. We will add a few sentences to clarify the reason for the high measurement uncertainty for Apatite-McClure in the manuscript.

- **RC 10: Figure 2:** For BaF-P2 and P4, some data points lie within 40 μm of the rim, contradicting the minimum distance limitation from the rim. How does this affect model results for these profiles (see comment line 178)? Does that be responsible for the lack of acceptable paths?

- **AR 10:** We have indeed measured $^4$He at a distance of less than 40 μm from the grain rim (see AR 8).
  As the reviewer points out in RC 8, excluding spots less than 40 μm from the rims may result in the loss of crucial information. However, in terms of inverse modelling (which aims to minimise the misfit between predicted and observed data), having fewer constraints or data points to fit should make it easier for the model to find acceptable paths rather than harder. We thus do not think having to exclude spots (e.g. for being too close to the rim for $C_{aw}$ calculation) is the reason for the lack of acceptable paths in Ap-BaF-P2 to Ap-BaF-P4.
  We observe that the un-invertible profiles Ap-BaF-P2 and Ap-BaF-P4 are significantly skewed and lack c-axis symmetry, while our inverse model is designed to operate under the assumption of c-axis symmetry. (The model is optimised for homogeneous grains.) The fact that the c-axis symmetric $^4$He profile Ap-BaF-P1 could be successfully inverted suggests to us that the model design and the assumption of symmetry are critical factors. We will include a few additional sentences in Section 4.7 to clarify this point.

- **RC 11: Line 332:** The McClure 4He profile is not shown. Should this be included in supplementary materials?

- **AR 11:** We will remove sentences concerning Apatite-McClure from this section as they are not a main result.

- **RC 12: Line 334:** Spot size uncertainties (10-15%) appear inconsistent with volume errors reported in Table 3 (4-9%). Since pit volume depends on spot size, the propagated uncertainties should scale accordingly. Please clarify.

- **AR 12:** Thank you for bringing this to our attention. "Spot measurement uncertainties" refer to the $^4$He measurement uncertainties, not to He ablation pit dimensions or ablation spot size. The wording in the sentence is misleading and will be rephrased.

- **RC 13: Line 335:** The text seems to conflate 4He measurement error and spot measurement error; numbers also differ from Table 2. Please clarify what each percentage refers to.

- **AR 13:** We will re-phrase these sentences as per our response to RC 12.

- **RC 14: Line 376:** The blank rejection criterion requires signals >3x blank according to the text. However, URG ablation pit data appear to fall below this threshold. For instance, for the Ap-URG-1 spot, the calculated 4He signal is 2.73×107 atoms (using an apatite density of 3.2×10-12 g/μm³), whereas the average blank reported by the

authors is $1.08 \times 10^7$ atoms (or 0.0004 cc). Comparing the net 4He signal (after blank subtraction) with the blank value indicates that the measured signal is less than three times the blank. This relationship holds for all URG spots listed in Table 3, where signals are only slightly above twice the blank level. I suggest that the authors verify these calculations and discuss how this may affect the reported 4He uncertainties for the URG apatite.

- **AR 14:** We acknowledge that this is poorly worded in the text. We will revise it. For our measurement setup, we consider a signal significantly above 3x blank level and an SD <5% as ideal. However, we do not treat this as a strict rejection criterion. The reviewer is correct in their assessment that $^4$He concentrations for Apatite-URG are less than 3x the blank level. We consider this still acceptable, but not ideal. Generally, the lower the $^4$He signal and the closer to the blank level, the larger the associated measurement uncertainties. We report the $^4$He measurement quality (whether the $^4$He signal is >3x blank level or lower) in the supplementary information. For detailed equations, please refer to AR 6.

- **RC 15:** **Line 387:** What does "B1" refer to in Table 2?
- **AR 15:** This should read Table 2 and Table B1. We will fix this.

- **RC 16:** **Lines 422 - section 4.4:** The authors report an interesting and somewhat unexpected result in their modelled ages: the crystal cores yield younger ages, while the rims show older ages. This pattern is indeed counterintuitive. The authors suggest that this discrepancy might stem from the material removed during re-polishing between the 4He and parent nuclide analyses. However, it is unclear why they did not acquire additional data from the same layer as the 4He measurements, either directly within the initial laser spots (as done by Pickering et al., 2020) or from nearby locations. Such data could have provided a valuable cross-check to support the modelled Caw values. Because the thickness of the polished layer is not specified, it is possible that too much material was removed, potentially invalidating the 3D extrapolation at the actual 4He spot depth. A quick verification at or near the 4He ablation sites would have allowed the authors to confirm or adjust their model accordingly. As it stands, the observed age pattern is difficult to interpret and may indicate underlying issues with either the dataset or the modeling approach.
- **AR 16:** We kindly refer to RC 2/ AR 2 for the reasoning behind the re-polishing and the amount of material removed. We are unsure how measuring "directly within the initial laser spots (as done by Pickering et al., 2020)" as the reviewer suggests, would yield better information compared to our approach. Measuring "in" the ablation spot also means accessing information from below the $^4$He ablation layer. This is nicely illustrated in Fig. 4 of Pickering et al. (2020). In our opinion, we do essentially retrieve information from the same layer as we would when ablating inside the $^4$He pits.
  Further, in section 4.4, we initially aimed to highlight that the grinding and polishing required to expose the internal grain surface after Teflon mounting result in the loss of

approximately half of the grain and thus lead to a significant loss of information for heterogeneous grains. We will clarify in section 4.4 which specific polishing step we are referring to.

- **RC 17: Line 482-483:** The comparison presented for the BaF model (Fig. 7b) would have been more informative if the modelled curves had included a zoned parent nuclide distribution rather than only a homogeneous system. As the authors correctly point out, a homogeneous model does not adequately represent this grain's characteristics. Incorporating a modelled zoned profile based on the observed 2D parent nuclide distribution would have provided a more meaningful test of whether the Caw correction significantly improves the model outcomes, since its benefit is not particularly evident in the homogeneous case.
- **AR 17:** Thank you for this good suggestion. We will add such a profile to Fig. 7b.

- **RC 18: Figure 7:** Why is the 5x5 µm grid model for URG not shown? Additionally, since increasing grid resolution seems to improve model outputs, it would be helpful to test finer resolutions (e.g., 1 µm) and discuss limiting factors.
  **AR 18:** We do not show a 5x5 µm grid model for Apatite-URG in Fig. 7, because we did not calculate one. We agree with the reviewer that a higher grid resolution appears to improve the model outputs. However, as Fig. 7a shows, the 10x10 µm grid model is already indistinguishable from the homogenous model within measurement uncertainty, and the misfits between modelled and measured $^4$He profiles are nearly the same. Hence, we do not see justification for generating even finer resolution maps for Apatite-URG. We will add this reasoning for the missing 5x5 µm grid model for Apatite-URG to the manuscript.
  Regarding the general testing of finer interpolation resolutions and their limiting factors, in our opinion, this should be addressed in a separate study. It is important to not only systematically test the interpolation resolution for parent nuclide maps, but also the combination of ablation spot size and interpolation grid resolution. This is beyond the scope of the manuscript.

- **RC 19: Line 510-512:** The authors claim that six laser spots are sufficient to reconstruct a simple, rapid cooling history with minimal or no parent nuclide zonation. However, for more complex and strongly zoned samples, it appears that only five laser spots (e.g., Ap-BaF-P1) were enough to produce acceptable time–temperature paths. I find this conclusion somewhat surprising and possibly overinterpreted. Could the authors clarify how five laser spots were sufficient in the more complex case, and whether this is supported by model sensitivity tests?
- **AR 19:** We will modify the text to clarify that the minimum number of spots needed for a $^4$He profile is not strictly implied by our results. We will also add a paraphrased version of our explanation below to the manuscript.
  Generally, we recommend using as many points as possible to measure a $^4$He profile.

As the reviewer points out, we cannot derive the minimum number of spots in a [4]He profile purely based on our measurements. The number that we give is our best estimate based on the data that we have available and a mathematical thought experiment: The minimum number of [4]He spots required to define a [4]He profile can be compared to determining how many points are needed to define a curve. It is trivial that two points define a straight line, and given any two distinct points, there is only one unique line that can be drawn through them. The minimum number of points required to define a curve is a more nuanced problem, as it depends on the complexity of the curve. The simplest assumption is that if the definition of a straight line requires two points, a simple curve should require at least three points. Indeed, it can be shown that a simple quadratic function of the form $f(x) = ax^2 + bx + c$ is uniquely determined by three distinct points.

If we assume, in the simplest mathematical case, that the [4]He profile within a homogeneous grain has a parabolic shape, we would thus only need three points from the core to the rim to measure the [4]He profile. (We cannot make the same assumption for rim-to-rim profiles, because three points would mean defining one shared central point and one outer point for each half of the grain. This would result in describing a line in each grain half, rather than capturing a curved profile.)

For rim-rim profiles, we would double the number required for a core-rim profile, resulting in six spots. In the case that the [4]He profile is parabolic in shape and symmetric about the c-axis (meaning the vertex of the parabola is located at the grain centre), five spots can be sufficient. Then, the middle spot should be located exactly at the grain centre. By chance, Ap-BaF-P1 is sufficiently close to that condition, and five spots are enough for measuring the [4]He profile.

We do not know beforehand how complex the [4]He profile's shape in the grain will be, and thus how many measurements are needed. It can very well be more than six. However, we can conclude with some confidence that one should not start with less than six (rim-rim).

- **RC 20: Line 519-521:** Do the blue, red, and black curves shown in Fig. 7 represent two separate models extending from the core to the rim on each side (i.e., toward positive and negative distances from the center)? If so, this could explain the slight discrepancies observed between the curves at the grain center, where the two models meet. This point is not enough clearly stated in the text to my opinion, and it would be important to clearly specify that these are half-grain models (core-to-rim) rather than full-grain simulations (rim to rim). Is the same modeling approach also applied to the URG apatite? Similar minor mismatches are visible near the center of those plots as well.

- **AR 20:** We indeed merge two core-rim profiles for forward-modelling. We will add explanations to the text and modify the figure captions to highlight this.

- **RC 21: Appendix B:** This table provides information for the McClure sample but does not include the calculated in situ AHe date. Although this sample was not used in

the modeling section, the authors successfully measured 4He signals and corresponding Caw U-Th-Sm values for numerous ablation pits. Do these individual ages combine to yield a whole-grain equivalent age given at ~523 Ma (see table 1)? it would be useful to include calculated in situ AHe dates, as it provides additional context and comparability with URG, especially given similar parent nuclide homogeneity (Fig. 3).

- **AR 21:** We did not calculate AHe dates because we were doubtful of their significance, given the large uncertainties on the [4]He concentrations. It is an interesting question how the in situ AHe dates compare to the crystallisation age reported for the McClure Mountain Syenite. Previous studies (e.g., Anderson et al., 2017) report AHe dates that are significantly younger than the crystallisation age of ~523 Ma, ranging from ~100 to 140 Ma. We will add AHe dates for Apatite-McClure to the Appendix.

**References**

Danišík, M., McInnes, B. I. A., Kirkland, C. L., McDonald, B. J., Evans, N. J., and Becker, T. 2017: Seeing is believing: Visualization of He distribution in zircon and implications for thermal history reconstruction on single crystals, Science Advances, 3(2), 1–10, https://doi.org/10.1126/sciadv.1601121

N. Evans , J. Byrne , J. Keegan and L. Dotter LE , Determination of Uranium and Thorium in zircon, apatite, and fluorite: application to laser (U–Th)/He thermochronology, J. Anal. Chem., 2005, 10.1007/s10809-005-0260-1, 60, 1159 -1165.

Glotzbach, C. and Ehlers, T. A.: Interpreting cooling dates and histories from laser ablation in situ (U– Th–Sm)∕He thermochronometry: a modeling perspective, Geochronology, 6, 697–717, https://doi.org/10.5194/gchron-6-6972024 , 2024.

Pickering, J., Matthews, W., Enkelmann, E., Guest, B., Sykes, C., and Koblinger, B. M., 2020, Laser ablation (U-Th-Sm)/He dating of detrital apatite: Chemical Geology, v. 548, p. 119683. https://doi.org/10.1016/j.chemgeo.2020.119683

Phillips, D., and Matchan, E., 2013, Ultra-high precision 40Ar/39Ar ages for Fish Canyon Tuff and Alder Creek Rhyolite sanidine: new dating standards required?: Geochimica et Cosmochimica Acta, v. 121, p. 229-239. https://doi.org/10.1016/j.gca.2013.07.003

Reiners, P. W., Farley, K. A., and Hickes, H. J., 2002, He diffusion and (U-Th)/He thermochronometry of zircon: initial results from Fish Canyon Tuff and Gold Butte: Tectonophysics, v. 349, no. 1-4, p. 297- 308. https://doi.org/10.1016/S0040-1951(02)00058-6

Renne, P. R., Swisher, C. C., Deino, A. L., Karner, D. B., Owens, T. L., and DePaolo, D. J., 1998, Intercalibration of standards, absolute ages and uncertainties in 40Ar/39Ar dating: Chemical Geology, v. 145, no. 1-2, p. 117-152. https://doi.org/10.1016/S0009-2541(97)00159-9

Additional references cited by the authors in their response:

Anderson, A. J., Hodges, K. V., & van Soest, M. C. (2017). Empirical constraints on the effects of radiation damage on helium diffusion in zircon. *Geochimica et Cosmochimica Acta*, *218*, 308–322. https://doi.org/10.1016/j.gca.2017.09.006

Binder, T., Marks, M. A. W., Gerdes, A., Walter, B. F., Grimmer, J., Beranoaguirre, A., Wenzel, T., & Markl, G. (2023). Two distinct age groups of melilitites, foidites, and basanites from the southern Central European Volcanic Province reflect lithospheric heterogeneity. *International Journal of Earth Sciences*, *112*(3), 881–905. https://doi.org/10.1007/s00531-022-02278-y

Meesters, A. G. C. A., and T. J. Dunai (2005), A noniterative solution of the (U-Th)/He age equation, *Geochem. Geophys. Geosyst.*, 6, Q04002, doi:10.1029/2004GC000834.